# Pentavalent HIV-1 vaccine protects against simian-human immunodeficiency virus challenge

Todd Bradley[1], Justin Pollara[1], Sampa Santra[2], Nathan Vandergrift[1], Srivamshi Pittala[3], Chris Bailey-Kellogg[3], Xiaoying Shen[1], Robert Parks[1], Derrick Goodman[1], Amanda Eaton[1], Harikrishnan Balachandran[2], Linh V. Mach[2], Kevin O. Saunders[1], Joshua A. Weiner[3], Richard Scearce[1], Laura L. Sutherland[1], Sanjay Phogat[4], Jim Tartaglia[4], Steven G. Reed[5], Shiu-Lok Hu[6], James F. Theis[7], Abraham Pinter[7], David C. Montefiori[1], Thomas B. Kepler[8], Kristina K. Peachman[9,10], Mangala Rao[9], Nelson L. Michael[9], Todd J. Suscovich[11], Galit Alter[11], Margaret E. Ackerman[3], M. Anthony Moody[1], Hua-Xin Liao[1], Georgia Tomaras[1], Guido Ferrari[1], Bette T. Korber[12] & Barton F. Haynes[1]

The RV144 Thai trial HIV-1 vaccine of recombinant poxvirus (ALVAC) and recombinant HIV-1 gp120 subtype B/subtype E (B/E) proteins demonstrated 31% vaccine efficacy. Here we design an ALVAC/Pentavalent B/E/E/E/E vaccine to increase the diversity of gp120 motifs in the immunogen to elicit a broader antibody response and enhance protection. We find that immunization of rhesus macaques with the pentavalent vaccine results in protection of 55% of pentavalent-vaccine-immunized macaques from simian–human immunodeficiency virus (SHIV) challenge. Systems serology of the antibody responses identifies plasma antibody binding to HIV-infected cells, peak ADCC antibody titres, NK cell-mediated ADCC and antibody-mediated activation of MIP-1β in NK cells as the four immunological parameters that best predict decreased infection risk that are improved by the pentavalent vaccine. Thus inclusion of additional gp120 immunogens to a pox-prime/protein boost regimen can augment antibody responses and enhance protection from a SHIV challenge in rhesus macaques.

[1] Duke Human Vaccine Institute, Duke University Medical Center, Durham, North Carolina 27710, USA. [2] Beth Israel Deaconess Medical Center, Harvard Medical School, Boston, Massachusetts 02215, USA. [3] Dartmouth College, Hanover, New Hampshire 03755, USA. [4] Sanofi Pasteur, Swiftwater, Pennsylvania 18370, USA. [5] Infectious Disease Research Institute, Seattle, Washington 98109, USA. [6] Department of Pharmaceutics, University of Washington, Seattle, Washington 98195, USA. [7] Public Health Research Institute, New Jersey Medical School, Rutgers University, Newark, New Jersey 07103, USA. [8] Department of Microbiology, Boston University, Boston, Massachusetts 02215, USA. [9] US Military HIV Research Program, Walter Reed Army Institute of Research, Silver Spring, Maryland 20910, USA. [10] Henry M. Jackson Foundation for the Advancement of Military Medicine, Bethesda, Maryland 20817, USA. [11] Ragon Institute of MGH, MIT, and Harvard, Cambridge, Massachusetts 02139, USA. [12] Los Alamos National Laboratories, Los Alamos, New Mexico 87545, USA. Correspondence and requests for materials should be addressed to T.B. (email: todd.bradley@duke.edu) or to B.F.H. (email: barton.haynes@duke.edu).

An effective human immunodeficiency virus (HIV)-1 vaccine will need to protect against acquisition of infection. The RV144 phase III HIV-1 vaccine trial in Thailand (NCT00223080) demonstrated vaccine efficacy of 60.5% at 12 months (*post hoc* analysis) that waned to modest protective efficacy (31.2%) after 42 months and is the only human clinical trial that has demonstrated protection from HIV-1 acquisition[1,2]. The results of the trial were unexpected since previous trials in the United States and Thailand with related vaccine boost components failed to protect[3,4]. A major goal of HIV-1 vaccine design is to elicit broad and potent neutralizing antibodies (bnAbs), but bnAbs are rarely made during HIV infection, and to date, no vaccine has elicited bnAbs[5,6]. Efforts to define the correlates of reduced infection risk in RV144 revealed that IgG antibody responses to the HIV-1 envelope (Env) variable loop 1 and variable loop 2 (V1–V2) epitopes were associated with a lower risk of infection, as were antibody-dependent cell-mediated cytotoxicity (ADCC)-mediating IgG antibodies with low IgA antibodies against Env subunit gp120 (refs 7–10). Genetic analysis of breakthrough infections revealed that lysine (K) 169 in the V2 was a site of immune pressure, and vaccine efficacy was 48% against viral strains matching this residue[11]. Thus, increasing the magnitude and breadth of the V2 antibody response may be a strategy to improve the efficacy observed in RV144.

The Thai HIV-1 epidemic is dominated by circulating recombinant form CRF01_AE; however, subtype B and CRF01_AE/subtype B recombinants are also relatively common[12]. Most of the CRF01_AE Env is subtype E and subtype A in the rest of the virus[13]. The RV144 vaccine consisted of a canarypox-vectored (ALVAC-AE; vCP1521) prime, expressing Gag-Pro of HIV subtype B; a gp41 transmembrane-anchored subtype E gp120 Env (from the isolate 92TH023) and a bivalent combination of subtypes B (from isolate MN) and CRF01_AE (from isolate A244) gp120 (AIDSVAX B/E) alum-adjuvanted protein boost[1].

In an effort to improve the RV144 vaccine regimen, we performed a nonhuman primate (NHP) study to compare the protective efficacy of a pentavalent (subtypes B/E/E/E/E) gp120 protein boost to that of a RV144-like bivalent (B/E) gp120 protein boost following ALVAC-AE prime against a heterologous neutralization-resistant (Tier-2) simian–human immuno-deficiency virus (SHIV) challenge. The ALVAC-pentavalent (B/E/E/E/E) gp120 immunogen included expanded potential epitope variants in V2 and gp120 epitopes outside of the V2 loop relative to the ALVAC-bivalent (B/E). The heterologous challenge virus we used (termed SHIV-1157(QNE)Y173H) was a cloned virus derived from the CCR5-tropic Tier-2 subtype C SHIV1157ipd3N4 (ref. 14) with three mutations introduced relative to the parental SHIV to enhance V2 region antibody recognition. Thus vaccine mediation of decreased transmission risk seen in this study would be associated with new epitopes present in the pentavalent B/E/E/E/E vaccine. We determined the immune correlates of delayed infection risk to inform future pox-vectored prime, gp120 boost vaccine designs.

## Results

**Design of a pentavalent gp120 vaccine boost**. We analysed HIV-1 Env sequences from the RV144 trial, including 44 vaccine-breakthrough infections (infections in the vaccine arm), and viruses from 66 infected individuals from the placebo group, in order to select additional Envs that improve HIV Env sequence coverage in the Thai population and could complement the bivalent protein component of the RV144 vaccine. Knowing that there was evidence of protective immune pressure against the V2 region in RV144 (ref. 11), we optimized epitope coverage to this region rather than the whole Env. A244 and 92TH023, the two

subtype E Envs used in the RV144 trial, are identical in the V2 region. Both of these vaccine antigens were isolated early in the Thai epidemic (A244 in 1990, 92TH023 in 1992) at a point when the circulating viruses in Thailand had minimal divergence, but by the time of the RV144 trial the CRF01_AE virus in Thailand had diverged considerably (Supplementary Fig. 1A).

We analysed overlapping eight amino acid-long sequences spanning the V2 region using the mosaic design tool[15] and identified three natural Env sequences (AA058, AA104 and AA107) from those sampled in the RV144 trial, which in combination with A244 and 92TH023, provided the best coverage of the V2 epitope region across all the CRF01_AE RV144 samples (Fig. 1a, Supplementary Fig. 1A,B). While the three Envs were selected for optimal diversity coverage of linear epitopes in the V2 region, by including additional strains in the pentavalent protein boost we also substantially improved diversity of the full gp120 relative to the original RV144 bivalent boost (Fig. 1a). Not only is subtype E diversity represented by the RV144 trial samples better accounted for in the pentavalent vaccine, the pentavalent vaccine also substantially improved potential epitope coverage of the C subtype SHIV challenge virus used in this study in both the V2 and across the gp120 (Fig. 1a).

We then immunized two groups ($n = 9$ per group) of Indian-origin rhesus macaques (*Macaca mulatta*) six times with $10^8$ plaque-forming units of ALVAC-AE (vCP1521) at weeks 0, 4, 13, 21, 47 and 88 and boosted with a either bivalent (B.63521/AE.A244 gp120) Env protein (Group 1) or pentavalent (B.63521/AE.A244/AE.AA058/AE.AA104/AE.AA107 gp120) Env protein (Group 2) at weeks 13, 21, 47 and 88 in GLA-SE adjuvant (Fig. 1b and Supplementary Fig. 1C). The subtype B Env 63521 was used rather than subtype B MN that was present in the RV144 vaccine, since it induced robust V1V2 antibody responses in Rhesus macaques[16].

**Vaccine-protective efficacy against SHIV challenge**. To assess the protective efficacy of the vaccines, we challenged all animals beginning at week 90 (2 weeks after the last ALVAC + protein immunization) eight times with a weekly intrarectal low-dose heterologous Tier-2 SHIV-1157(QNE)Y173H. Eight unvacci-nated animals were challenged as the control arm. The V2 region of the challenge virus was mutated in three positions to optimally bind RV144 V2 antibodies CH58 and CH59 and V2 bnAbs PG9 and PG16 (Supplementary Fig. 1D). Single genome amplification of the challenge SHIV stock *env* sequences revealed that out of 56 *env* sequences only 5 had >3 nucleotide mutations, which indicated little sequence diversity of the challenge virus stock (Supplementary Fig. 1E). After eight challenges, 1 of the 9 (11.1%) bivalent (B/E) and 5 of the 9 (55.6%) pentavalent (B/E/E/E/E) vaccinated animals remained uninfected, indicating significant improvement in protection by the pentavalent vaccine (Fig. 1c; $P = 0.02$, Kaplan–Meier (KM) log-rank test). In the control arm, 2 of the 8 (25%) animals remained uninfected at week 11 postchallenge (Fig. 1c; group 2 versus group 3, $P = 0.15$, KM log-rank test). Statistical significance was not achieved when the pentavalent group was compared to control unvaccinated animals, but when the control and bivalent groups were com-bined after eight challenges, there was a significant difference between the combined group and the pentavalent group ($P = 0.03$, KM log-rank test). There was no significant difference between the vaccinated or control groups in peak viral load or control of viremia after infection (Fig. 1d,e). Thus the ALVAC-pentavalent HIV vaccine afforded significantly better protection against acquisition of infection following repetitive, intrarectal, SHIV-1157(QNE)Y173H challenges than did an ALVAC-bivalent vaccine regimen.

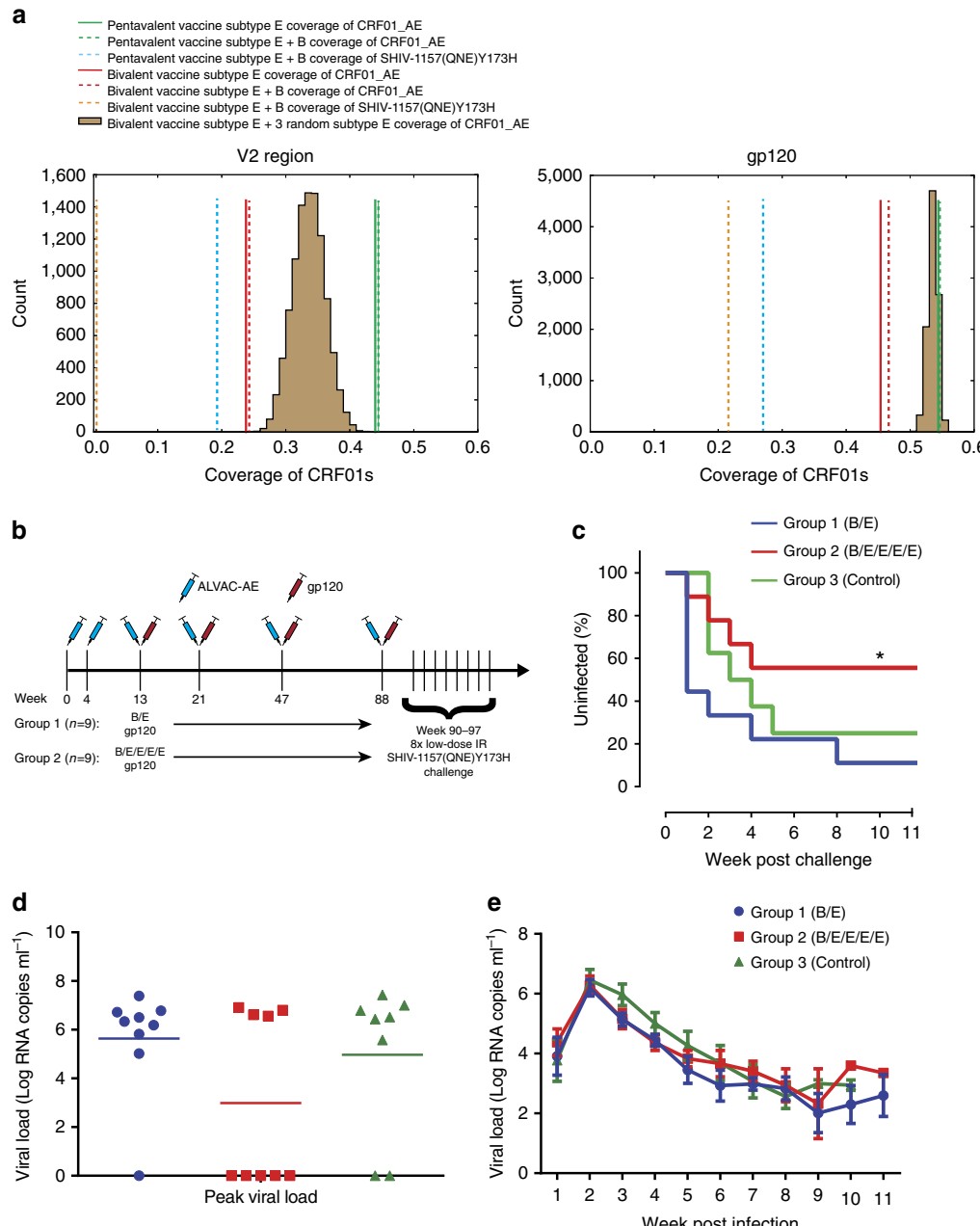

**Figure 1 | Pentavalent vaccine improved coverage of HIV-1 diversity and had increased protection from SHIV challenge. (a)** The sequence coverage of the RV144 viral V2 epitope region (HXB2 positions 154–184) and full subtype E gp120 by the pentavalent vaccine subtype Es (green line; 92TH023, A244, AA058, AA104 and AA107), the pentavalent vaccine subtype Es and B (63521; dashed green line), the pentavalent subtype E and B coverage of the challenge SHIV-1157(QNE)Y173H (dashed blue line), the RV144 bivalent subtype Es (92TH023 and A244; red line), the RV144 bivalent subtypes E and B (63521; dashed red line) and the RV144 bivalent vaccine subtype E plus B SHIV coverage (dashed orange line). The distribution of sequence coverage of 10,000 randomly selected sets of three clade E viruses when combined with A244 and 92TH023 (brown). **(b)** Schematic of the immunization and challenge regimen. Eighteen rhesus macaques are administered two doses of ALVAC-AE, and then animals either received ALVAC-AE plus a bivalent ($n = 9$) or pentavalent ($n = 9$) protein boost four times. Then all animals were subjected to 8 weekly low-dose intrarectal challenges with SHIV-1157(QNE)Y173H. Unimmunized animals ($n = 8$) were challenged as the control arm. **(c)** KM plot showing the percentage of uninfected animals after 8 weekly challenges (*Group 2 vs Group 1, $P = 0.02$; Group 2 vs Group 3, $P = 0.48$; one-tailed KM log-rank test). **(d)** Peak viral load of the infected animals from each vaccine and control group. **(e)** Viral load tested weekly after initial infection in all the groups. Lines are group means and error bars indicate s.e.m.

**Pentavalent vaccine increased binding antibody titres.** We next determined differences in antibody responses between the bivalent- and pentavalent-vaccinated monkey groups. Binding antibody responses against the vaccine Env proteins were not detected by enzyme-linked immunosorbent assay (ELISA) after the ALVAC prime but were detected 2 weeks after the first protein boost for both groups and increased with subsequent

boosts (Fig. 2a). There was a trend of higher antibody-binding titres observed prechallenge (week 90) for the pentavalent-immunized animals against B.63521, A244 and AA058 gp120 proteins used as vaccine immunogens and significantly increased binding titres for AA104 and AA107 proteins, two of the Envs only present in the pentavalent vaccine ($P = 0.01$ for AA104 and $P = 0.008$ for AA107, Wilcoxon–Mann–Whitney test; Fig. 2b).

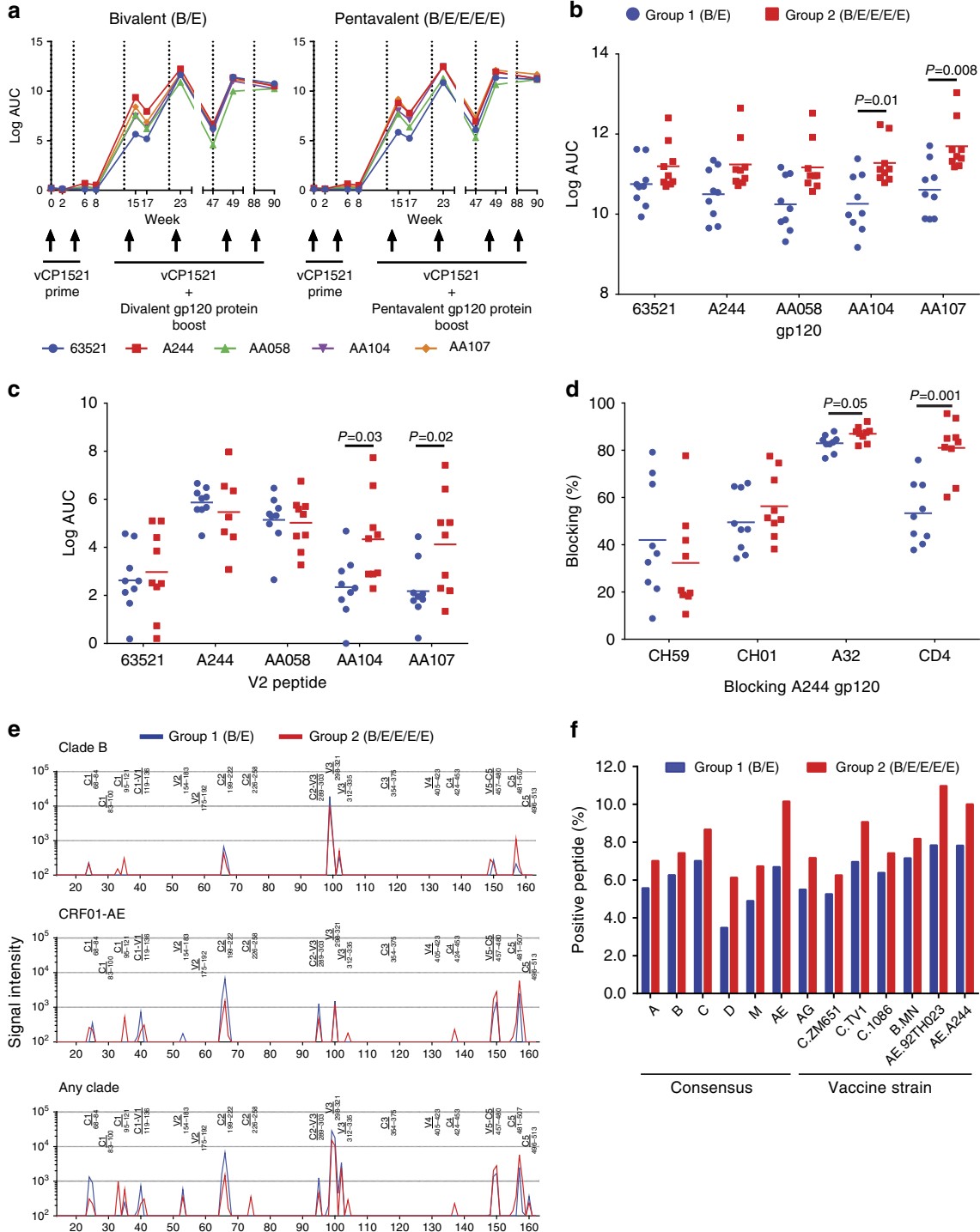

**Figure 2 | Binding antibody responses of ALVAC-bivalent and ALVAC-pentavalent vaccines.** (**a**) ELISA binding of plasma antibodies to the five vaccine Envs over the course of immunization for the bivalent (left) and pentavalent (right) immunized animals. Binding titres measured as mean Log area under curve (Log AUC) starting at a 1:30 plasma dilution. (**b,c**) Plasma antibody binding prechallenge at week 90 to (**b**) vaccine gp120s and (**c**) vaccine V2 peptides by ELISA measured as Log AUC. Horizontal bars are the group mean and P values calculated using a Wilcoxon–Mann–Whitney test. (**d**) Plasma antibody blocking prechallenge at week 90 of CH59, CH01 and A32 antibody and sCD4 binding to A244 gp120 measured as percentage of blocking. P values calculated using Wilcoxon–Mann–Whitney test. (**e**) Graph of binding signal intensity for peptides over the HIV-1 Env from clades B and CRF01-AE and any clade for the bivalent (blue) and pentavalent (red) immunization groups of plasma IgG using a peptide microarray. Peptide microarray contains overlapping 15-mer peptides covering the Env gp160. HIV-1 Env regions labelled above lines. Plotted are the group median values normalized for IgG concentration. (**f**) Bar graph showing the percentage of positive peptides in a peptide microarray that contains overlapping 15-mer peptides covering the Env gp160 for seven clade consensus (A, B, C, D, M, AE, AG) and six vaccine strains (ZM651, TV1, 1086, MN, 92TH023 and A244) for all animals in both vaccine groups. Group average reported. Positive threshold is signal intensity/IgG concentration >100.

We next tested both vaccine groups for antibody binding to V2 peptide regions from all five vaccine Envs. Equivalent levels of binding antibody were observed for 63521, A244 and AA058 in the two groups; however, significantly higher antibody titres were detected for the pentavalent-immunized animals against AA104 and AA107 V2 regions ($P = 0.05$ for AA104 and $P = 0.001$ for AA107, Wilcoxon–Mann–Whitney test; Fig. 2c). Using competitive ELISA, we observed similar levels of antibodies that could block CH59, a V2-targeting antibody isolated from an RV144 subject that mediates ADCC[17], and CH01, a bnAb that targets the V1V2-glycan site[18], for both vaccine groups (Fig. 2d). The pentavalent-vaccinated animals had higher levels of antibodies that blocked monoclonal antibody (mAb) A32 (ADCC-mediating antibody isolated from an infected individual[19]) binding to Env ($P = 0.05$; Wilcoxon–Mann–Whitney test) and binding of soluble CD4 (The primary cell receptor for HIV) to the Env ($P = 0.001$;

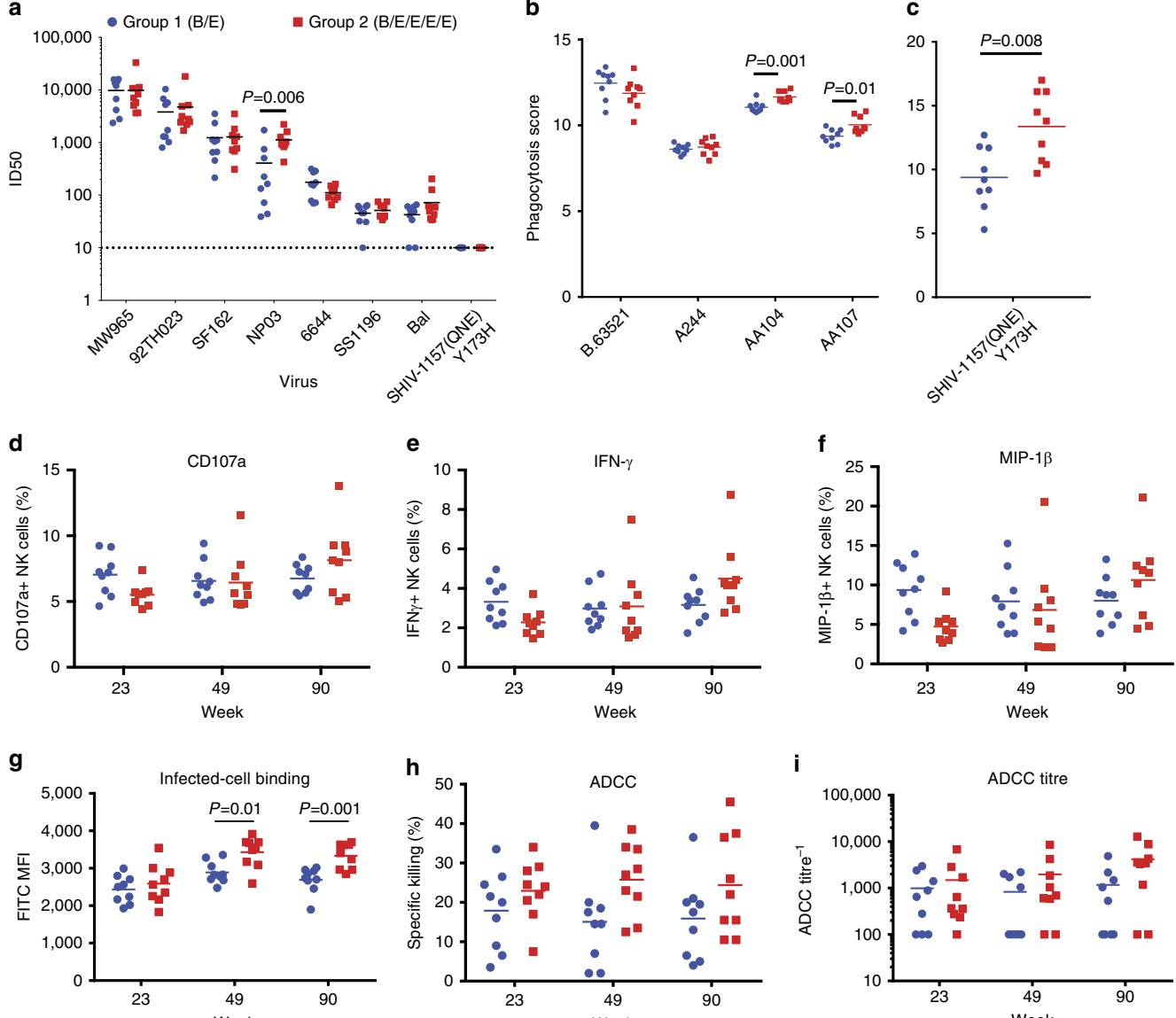

**Figure 3 | Plasma antibody neutralizing and non-neutralizing effector functions.** (**a**) Plasma neutralization of Tier-1 viruses and the Tier-2 challenge SHIV prior to challenge at week 90 measured in the TZM-bl neutralization assay. ID50 of individual animals displayed. Horizontal bars are the group mean. *P* values calculated using Wilcoxon–Mann–Whitney test. (**b,c**) Phagocytosis of vaccine Env and SHIV-1157(QNE)Y173 gp120-coated beads by group 1 and group 2 plasma before challenge at week 90 by THP-1 cells. Palivizumab and HIVIG used as negative and positive control antibodies. Average of two replicate experiments. Bead phagocytosis was quantified using the phagocytosis score. Horizontal bars are the group mean. *P* values calculated using Wilcoxon–Mann–Whitney test. Panel ADCP assay was performed using a lower cell density to increase assay sensitivity. (**d–f**) Antibody-dependent NK cell activation. Purified plasma IgG from each animal from groups 1 and 2 were tested for surface expression or production of (**d**) CD107a, (**e**) intracellular expression of IFN-γ (**f**) and MIP-1β by primary NK cells in the presence of A244 gp120. Average of two replicate experiments. Horizontal bars are the group mean. (**g**) Plasma antibody binding from group 1 (B/E) in blue and group 2 (B/E/E/E/E) in red against CM235-infected CD4 T cells (CEM.NKR$_{CCR5}$) measured after the last three protein boosts at weeks 23, 49 and 90 using the Luc-based ADCC assay. Horizontal bars are the group mean. *P* values calculated using Wilcoxon–Mann–Whitney test. (**h**) Plasma antibody ADCC from group 1 (B/E) in blue and group 2 (B/E/E/E/E) in red of CM235-infected CD4 T cells (CEM.NKR$_{CCR5}$) measured by percentage of cell killing after the last three protein boosts at weeks 23, 49 and 90. (**i**) Peak ADCC antibody titres (end point plasma dilution above previous established positive cutoff) from group 1 (B/E) in blue and group 2 (B/E/E/E/E) in red of CM235-infected CD4 T cells (CEM.NKR$_{CCR5}$) after the last three protein boosts at weeks 23, 49 and 90. Horizontal bars are the group mean.

Wilcoxon–Mann–Whitney test; Fig. 2d). Plasma-blocking antibodies were induced at high titres after the second B/E or B/E/E/E/E protein boost and increased with subsequent immunizations (Supplementary Fig. 2A).

Next we characterized plasma IgG antibody-binding responses at week 90 to HIV-1 linear epitopes covering the full gp160 of HIV-1 Env from seven consensus clades and circulating recombinant forms (A, B, C, D, group M, CRF01_AE and CRF02_AG) and six multi-clade vaccine strains. Linear epitopes within the Env variable loop 3 (V3) region were the dominant response for both immunization groups against clade B and clade C strains, but there were additional IgG-binding antibodies that targeted the C2, V5–C5 and C5 regions in the CRF01-AE clade (Fig. 2e, Supplementary Fig. 2B,D). Both vaccine groups had similar antibody-binding patterns and intensities against linear epitopes across the HIV Env, but the pentavalent group did have more detectable binding to a peptide region in the C1 for all clades tested (Fig. 2e). The pentavalent vaccine group had a higher average percentage of positive peptides for both the consensus and vaccine HIV linear epitopes, but differences did not achieve statistical significance (Fig. 2f and Supplementary Fig. 2C). These results indicated that the both the bivalent and pentavalent vaccines elicited a similar profile of binding antibodies to the HIV-1 Env, but the pentavalent-immunized animals had improved antibody-binding titres to select Envs present only in the pentavalent vaccine, including antibodies that targeted the V2 and CD4-binding site (CD4bs).

**Both vaccines elicited mucosal antibody responses**. The RV144 vaccine trial consisted of two ALVAC-AE primes followed by two ALVAC + protein boosts, and while systemic antibody levels were determined, analysis of mucosal tissue was not performed. We performed binding antibody multiplex assays for 12 multi-clade HIV Env proteins from mucosal IgG antibodies isolated from rectal wecks taken after the second protein boost at week 23. We found that all animals had detectable IgG antibodies in the rectal mucosal at week 23 against at least one of the tested antigens. Moreover, there was no significant difference in antibody-binding titres between the vaccine groups (Supplementary Fig. 3). These results demonstrate that both vaccines elicited robust mucosal IgG antibody responses.

**Vaccine-induced antibodies had multiple effector functions**. Neutralizing antibodies (nAbs) against seven neutralization-sensitive Tier-1 viruses were detected by the TZM-bl virus neutralization assay at week 90 prechallenge, but no neutralization was observed for the neutralization-resistant Tier-2 challenge SHIV (Fig. 3a and Supplementary Fig. 4A). Both vaccine groups neutralized six of the seven Tier-1 viruses equally, but the pentavalent group had significantly higher neutralization titres against the subtype E NP03 Tier-1 virus ($P = 0.006$, Wilcoxon–Mann–Whitney test).

Both vaccine groups had prechallenge antibody titres that could mediate phagocytosis (antibody-dependent cellular phagocytosis (ADCP)) of vaccine Env-coated targets (Fig. 3b,c and Supplementary Fig. 4B). Equivalent phagocytosis of B.63521 and A244 was observed for group 1 and group 2, but there was significantly higher phagocytosis of AA104- and AA107-coated beads for the pentavalent-immunized group ($P = 0.001$ and $P = 0.01$, respectively, Wilcoxon–Mann–Whitney; Fig. 3b). Using a lower cell density to increase assay sensitivity was required to accurately detect differences in ADCP of the challenge SHIV. There was significantly higher phagocytosis of the SHIV-1157(QNE)Y173H Env-coated beads ($P = 0.008$,

Wilcoxon–Mann–Whitney; Fig. 3c). These data suggest that vaccine-induced IgG was functional for engaging monocytes.

Antibody-dependent complement deposition (ADCD) was assayed by measuring the deposition of complement component C3b on the surface of CD4-expressing target cells coated with A244 gp120. Both vaccine groups had detectable ADCD after the second protein boost (week 23) where the pentavalent vaccine trended higher, but prechallenge at week 90 both vaccine groups had nearly 100% C3b deposition indicating high levels of ADCD (Supplementary Fig. 4C). Similarly, antibody-dependent neutrophil-dependent phagocytosis of A244-coated targets was detected of both vaccine groups (Supplementary Fig. 4D). Antibody-dependent natural killer (NK) cell activation was detected by the surface expression of CD107a and intracellular production of interferon (IFN)-γ and macrophage-inflammatory protein (MIP)-1β in NK cells that had been incubated with plasma antibodies from both vaccine groups and A244 gp120-coated CEM-NKr cells after the last three protein boosts. Both vaccine groups had antibody-dependent NK cell activation with the pentavalent vaccine group averaging higher for CD107a, IFN-γ and MIP-1β NK cells prechallenge (Fig. 3d–f).

Non-neutralizing functional ADCC antibody levels were tested using Tier-2 clade AE CM235 virus-infected CD4$^+$ T cells for both vaccine groups before immunization and after the last three protein boosts. As measured by mean fluorescence intensity (MFI), pentavalent-immunized animals had higher titres of antibodies that bound to the surface of virus-infected CD4$^+$ T cells at weeks 49 and 90 ($P = 0.01$ and $P = 0.001$, respectively; Wilcoxon–Mann–Whitney test; Fig. 3g). Pentavalent-immunized animals trended higher for peak ADCC at all three time points but did not reach statistical significance (Fig. 3h). Similarly, the ADCC titres at peak cell killing for the pentavalent-immunized animals averaged higher at weeks 49 and 90 (Fig. 3i). Both bivalent- and pentavalent-immunized animals had ADCC activity against A244, AA104, AA107 and SHIV-1157(QNE)Y173H gp120-coated target cells but there was no difference in peak ADCC levels or antibody titres between the two groups (Supplementary Fig. 4E). These data demonstrate that both vaccine groups lacked neutralization of the Tier-2 challenge SHIV before challenge, but the pentavalent group had improved recognition of Tier-2-infected CD4 T cells and ADCP.

**Identification of vaccine-specific antibody signatures**. To broadly profile the vaccine-induced polyclonal antibody response, we utilized a systems serology approach integrating multiple diverse prechallenge measurements. Plasma antibody binding (11 measures), blocking (13 measures), neutralization (7 measures), ADCP (5 measures), NK cell surface marker expression (3 measures) and ADCC (13 measures) were all assessed at the time of challenge (week 90) (Supplementary Fig. 5A). In addition, Fc characteristics of antigen-specific antibodies (36 antigens, 13 Fc-receptor detection reagents) were assessed using a customized Fc array assay, and 6 additional Fc-effector functions (ADCC, ADCP, ADCD and three Ab-dependent NK cell activities) were assayed at 6 time points throughout the immunization regimen (Supplementary Fig. 5B,C).

We first sought to profile differences in humoral responses between the two vaccinated groups using a logistic regression classifier to identify linear combinations of antibody and functional activity measurements that robustly distinguish the groups. Cross-validated classifiers trained on measurements taken after the initial boost (week 15), at which point the two vaccine regimens first diverged, achieved near-perfect discrimination between the two groups (Supplementary Fig. 5D). Perfect discrimination between groups was observed after the second

boost (week 23) (Fig. 4a) as well as for the remainder of the study (Supplementary Fig. 5D), and repeated cross-validation and permutation testing confirmed the robustness of this classification approach (Supplementary Fig. 5E). The classifiers identified measurements corresponding to antibody activity against immunogens present only in the pentavalent vaccine (AA058, AA104 and AA107) as those crucial for attaining perfect classification of the animals (final model shown in Fig. 4b,c). The pentavalent boosting regimen rapidly elicited a broader response against Env antigenic variants, and over the course of the vaccination series, these measurements demonstrated evolving antibody-binding differences between the two groups (Fig. 4d and Supplementary Fig. 5F).

**Immune correlates of decreased infection risk.** To identify combinations of antibody and functional activity measurements predictive of vaccine trial outcome, we performed multivariate survival analysis with Cox proportional hazard (PH) models trained on a small set of informative, non-redundant variables. These prechallenge antibody measurements predicted probability of infection at each challenge time point, with group-wise predictions tracking closely to the observed the KM curves (Fig. 5a). Furthermore, on a per animal basis, predicted risk of infection was highly concordant with observed time to

infection (cross-validated Concordance index (C-index): 0.85; $P = 1.09$e-09) (Fig. 5b). Additionally, the animals in the pentavalent vaccine group had significantly lower predicted risk of infection than those in the bivalent vaccine group ($P = 2.76$e-03 Wilcoxon–Mann–Whitney) (Fig. 5c). Repeated cross-validation and permutation testing established confidence in the reliability of these results (Fig. 5d). This risk analysis uncovered four complementary correlates of decreased risk of infection: three measuring ADCC (antibody binding to HIV-infected cells, peak ADCC antibody titre and NK cell-mediated ADCC) and one measuring antibody-mediated activation of NK cells by MIP-1β intracellular expression (Fig. 5e and Supplementary Fig. 6A). Thus multivariate analysis enabled the discovery of a combination of prechallenge immunological parameters that captured both the group- and animal-level differences in the risk of infection.

We next evaluated the potential for the individual prechallenge immunological parameters to correlate with the number of challenges required to establish infection in both immunized groups (Supplementary Fig. 5A). Delayed infection was most correlated with ADCP of the challenge SHIV, but prechallenge levels of peak ADCC titres, Env binding, CD4 and CH01 blocking and infected-cell binding all significantly correlated with time to infection ($P < 0.05$; Spearman rank correlation; Fig. 5f). Related antibody parameters also correlated with delayed infection

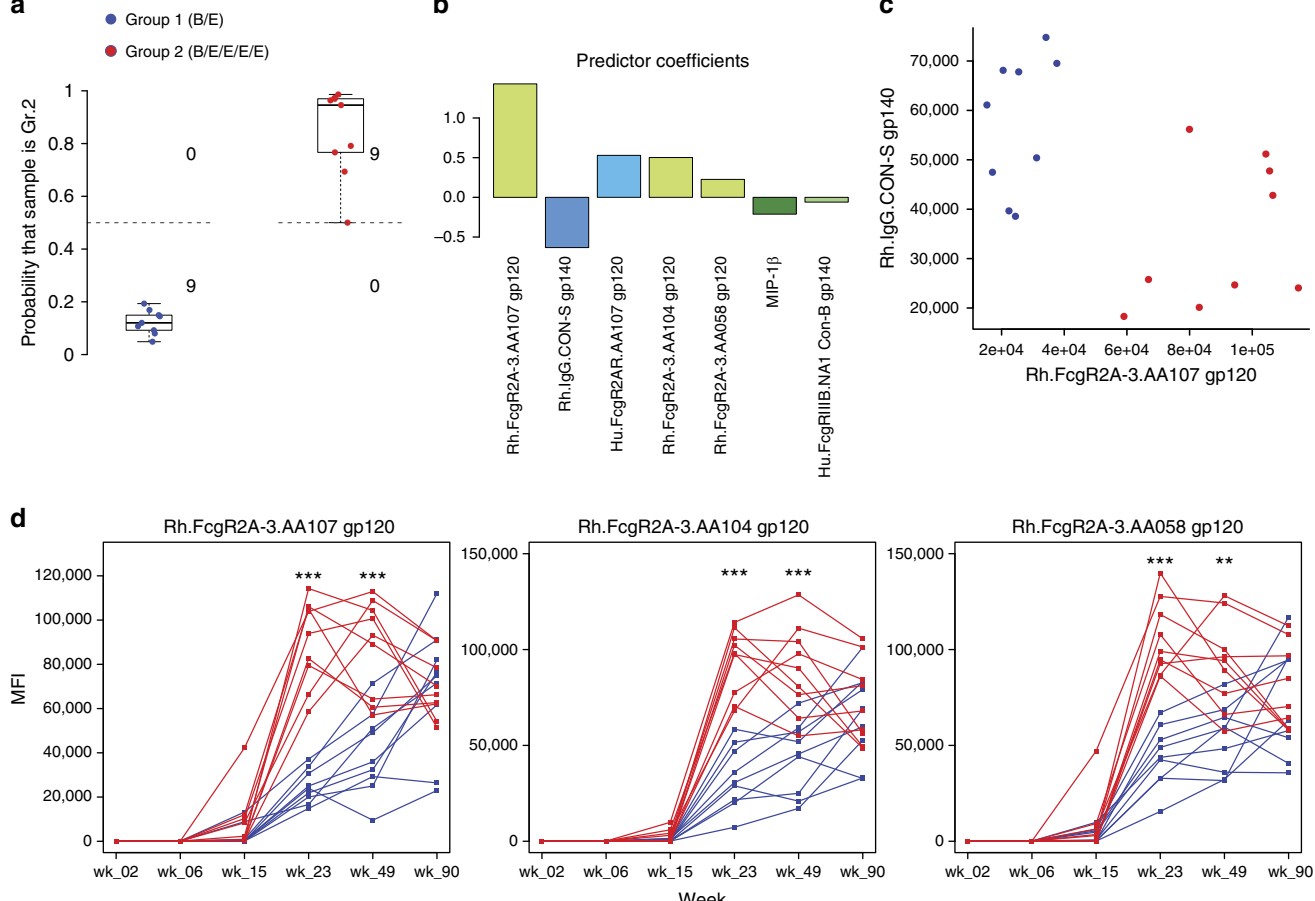

**Figure 4 | Bivalent and pentavalent vaccines elicited different antibody responses that evolved over time.** (**a**) Probability of each sample to be predicted as belonging to the pentavalent group. The decision boundary of the logistic classifier is shown at 0.5 (dashed line). (**b**) Logistic regression coefficients for the variables selected by the final LASSO-regularized model trained using all samples. The variables are ordered in descending order by magnitude of coefficient. (**c**) Bi-plot of the two variables with the highest magnitude coefficients in the final model. Each point represents a sample. (**d**) Temporal plots of variables correlated with the pentavalent group illustrating the evolution of group differences over time. The differences between groups, at each time point, were tested for significance with Wilcoxon–Mann–Whitney; *$P < 0.05$, **$P < 0.01$, ***$P < 0.001$.

(Supplementary Fig. 6B). A combined analysis of all the measures that had a significant Spearman correlation with delayed infection revealed that the pentavalent-immunized animals had a significantly better composite scores, indicating an overall better polyfunctional antibody response (Fig. 5g). Finally, we evaluated six prespecified parameters beginning with prechallenge samples

followed by weekly testing until infection to identify parameters significantly associated with protection using a Cox PH model. None of the parameters tested met statistical significance, most likely due to the small size of the study, but levels of AA107 Env-binding antibodies were associated with a 55.2% reduction of infection risk that neared the significance threshold

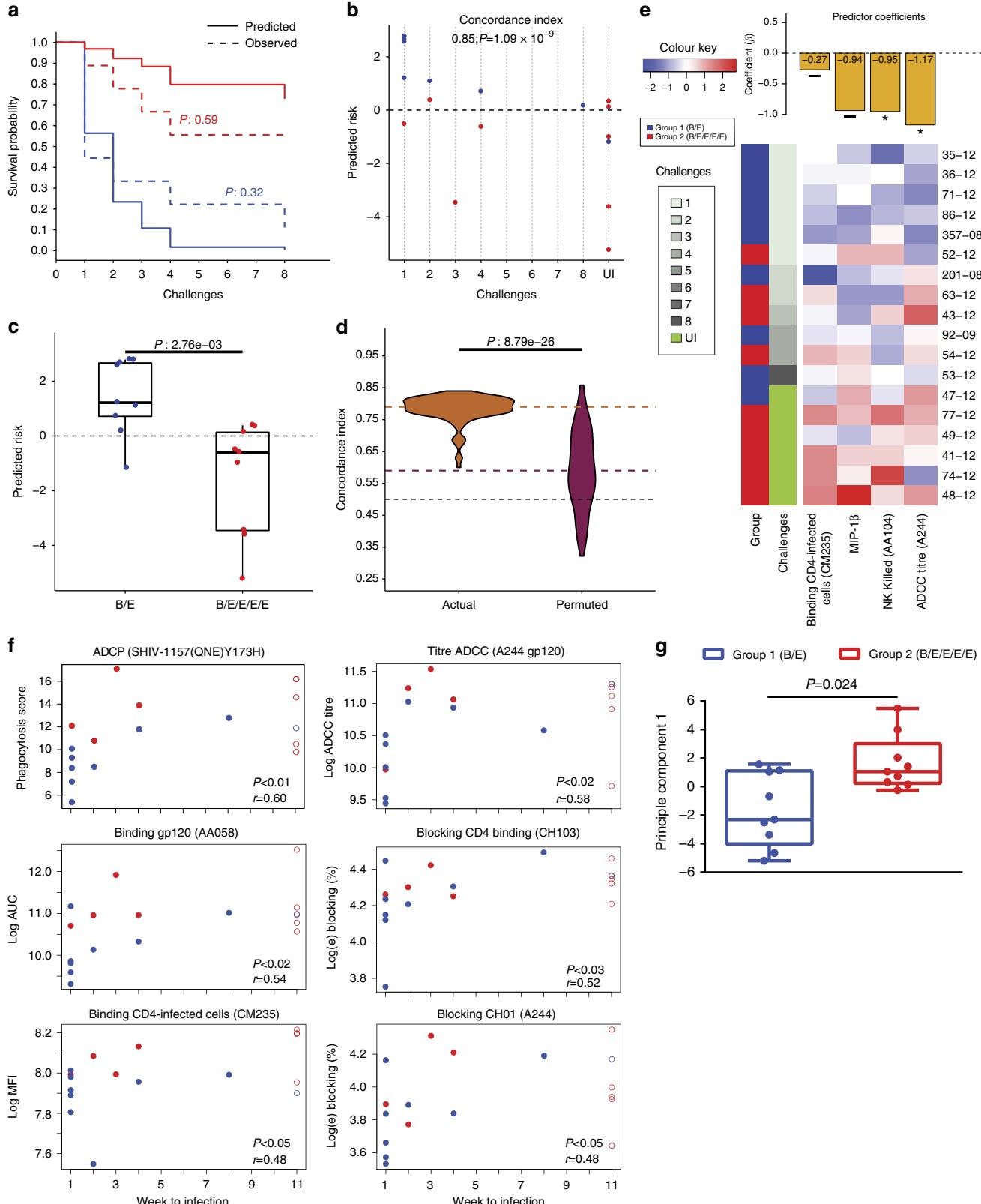

(hazard ratio = 0.448; $P = 0.0769$; Supplementary Fig. 6C). These data show that the pentavalent vaccine elicited a superior polyfunctional antibody response when compared to the bivalent vaccine that resulted in increased protection against SHIV challenge and immunological parameters associated with titres of binding antibodies and ADCC correlated with protection.

**The pentavalent vaccine-induced memory B-cell repertoire.** Increased antibody titres that blocked CD4 and mediate ADCC correlated with delayed infection. We sought to isolate antibodies from the pentavalent vaccine group that had these protective characteristics. HIV-reactive memory B cells were single-cell sorted using flow cytometry and immunoglobulin receptor genes amplified and sequenced from two animals that were immunized with the pentavalent vaccine at weeks 23 and 49 after the second and third protein boost, respectively (Supplementary Fig. 7A). Transient recombinant antibody expression and epitope mapping by ELISA identified 137 antibodies that reacted with HIV gp120 (Fig. 6a). Fifty-one of the antibodies were mapped to specific regions on the HIV Env, including the CD4bs, C1, V1/V2 and V2 linear epitope. Eight antibodies that target the CD4bs, gp120 epitope or the V1/V2 loop and had the broadest binding to the vaccine immunogens were selected for further study (Supplementary Fig. 7B). All but two antibodies, DH633 and DH640, reacted with all five vaccine Envs (Fig. 6b). Three V2-reactive antibodies (DH637, DH638 and DH641) that reacted strongly with all five vaccine Envs blocked the binding of 697D, a V2-specific conformation-dependent antibody isolated from an infected individual (Fig. 6c)[20]. All three antibodies failed to block the binding of CH58 and CH59, which target the V2 epitope that contains lysine (K) 169, indicating a V2 epitope distinct from that recognized by CH58 and CH59. Moreover, all three antibodies bound equally to A244 gp120 and A244 K169V mutant Env (Supplementary Fig. 7C). Additionally, DH637 and DH638 blocked the binding of the V2 bnAb CH01, suggesting overlapping epitopes with this bnAb (Fig. 6c). Epitope mapping of DH637 by linear microarray determined that it recognized linear peptides from diverse HIV-1 strains that contained the canonical $\alpha_4\beta_7$ gut mucosal homing receptor interacting site (LDV)[21] (Supplementary Fig. 7D). In addition, DH637 could inhibit the binding of $\alpha_4\beta_7$ on RPMI8866 cells to a cyclic form of 92TH023 (clade E Env) V2 peptide (Supplementary Fig. 7E). DH638, DH641 and a gp120-conformational mAb DH640 all mediated ADCC of CM235-infected CD4$^+$ T cells (Fig. 6d).

Four antibodies (DH631, DH632, DH633 and DH635) blocked the binding of CD4 to the Env and were sensitive to mutations within the CD4bs (Fig. 6e and Supplementary Fig. 7F). The four CD4bs, three V2 and gp120 antibodies were tested for neutralization. All eight antibodies neutralized Tier-1 viruses, but not the Tier-2 viruses Q168, CM244 and JRFL. Only the CD4bs

antibody DH635 showed weak neutralization of the Tier-2 challenge SHIV (Fig. 6f). DH637 had the broadest neutralization and DH640 only weakly neutralized 92TH023. Increased titres of antibodies that bind to Env, mediate ADCC, block CH01 and CD4 binding and NP03 neutralization all correlated with delayed time to infection and were higher in the pentavalent vaccine group (Fig. 5a). Thus the isolated antibodies from pentavalent-immunized animals have the characteristics of plasma antibodies that correlated with decreased infection risk.

## Discussion

In this study, we aimed to improve the breadth of responses in RV144 in order to improve vaccine-induced protection beyond that seen with the bivalent B/E vaccine used in Thailand.

The majority of HIV-1 vaccine candidates tested in rhesus macaques only demonstrated protection against neutralization-sensitive viruses and have failed to predict vaccine efficacy when translated into humans. Here we challenged the animals with a difficult-to-neutralize (Tier-2) challenge SHIV-1157(QNE)Y173H to determine whether protection could be observed in the absence of plasma antibody neutralization. We found that the ALVAC prime/ALVAC-pentavalent protein boost vaccine regimen afforded significant protection (55.6%) compared with no protection with an ALVAC-bivalent protein regimen. This suggests that epitope diversity in the additional Envs selected for optimal CRF01_AE coverage elicited a protective response in these animals. However, two of the eight control animals remained uninfected (25%), and the pentavalent vaccine did not reach statistical significance when compared to the control group alone due to the small number of animals in each group but was significant when the bivalent and control animals were grouped together. Future studies using pentavalent vaccine immunogens with larger numbers of animals will be required to fully determine the protective efficacy.

Both the bivalent and pentavalent vaccines elicited plasma and mucosal antibodies that bound to multiple epitopes on the HIV-1 Env that mediated neutralizing and non-neutralizing antibody functions, but no neutralization of the neutralization-resistant (Tier-2) challenge SHIV was observed for either vaccine group. Similar weak neutralization was observed in the RV144 trial and this indicated that other immune responses alone or in combination mediated the moderate protection that was observed[8]. An antibody that weakly neutralized the challenge SHIV, DH635, targeted the CD4bs and was isolated from the blood of a pentavalent-immunized macaque. It remains possible that low levels of these types of neutralizing antibodies at the rectal mucosal could have been involved in protection from infection.

Other vaccine regimens that mimic the RV144 protocol using SIV immunogens in rhesus macaques demonstrated partial

**Figure 5 | Immune correlates of decreased infection risk.** (**a**) Comparison between the group-wise observed KM curve and predicted survival probabilities. The probabilities were predicted for the 'mean animal' of each group, according to a final model trained with all animals. The $P$ values were calculated using the log-rank test. (**b**) Predicted relative risk of infection for individual animals over the representative ninefold cross-validation. The risk for each animal was predicted relative to the mean of all 18 animals, whose relative risk is 0 (horizontal dashed line). (**c**) Group-wise comparison of the predicted relative risk of infection for individual animals over the representative ninefold cross-validation. The $P$ value was calculated using Wilcoxon–Mann–Whitney. (**d**) Comparison of $C$-index values from 100 repetitions of ninefold cross-validation using actual (orange) versus permuted (red) time-to-infection labels. Significance was tested with Wilcoxon–Mann–Whitney. The horizontal dashed lines represent the median $C$-indices (actual labels: orange; permuted: red) and the baseline for random prediction (0.5: black). (**e**) Heatmap of the most predictive features from repeated cross-validation. The animals (rows) are ordered in ascending order of time-to-infection. The barplot at the top shows regression coefficients from the final model trained with all samples (Cox PH $P$ values: *$P < 0.05$; —, not significant). (**f**) Correlation of antibody ADCP, ADCC, binding and blocking at week 90 with the number of weeks after challenge required to establish infection. The plotted data reflect only the vaccinated animals and not controls. Group 1 (B/E) in blue and group 2 (B/E/E/E/E) in red; open circles indicate animals uninfected after eight challenges. $P$ values reflect Spearman rank correlation tests, and $r$ values reflect Spearman rho. (**g**) Graph of the PC composite score, which is the first principle component of the measures that had a significant Spearman correlation with weeks to infection. The group comparison was performed by Exact Wilcoxon.

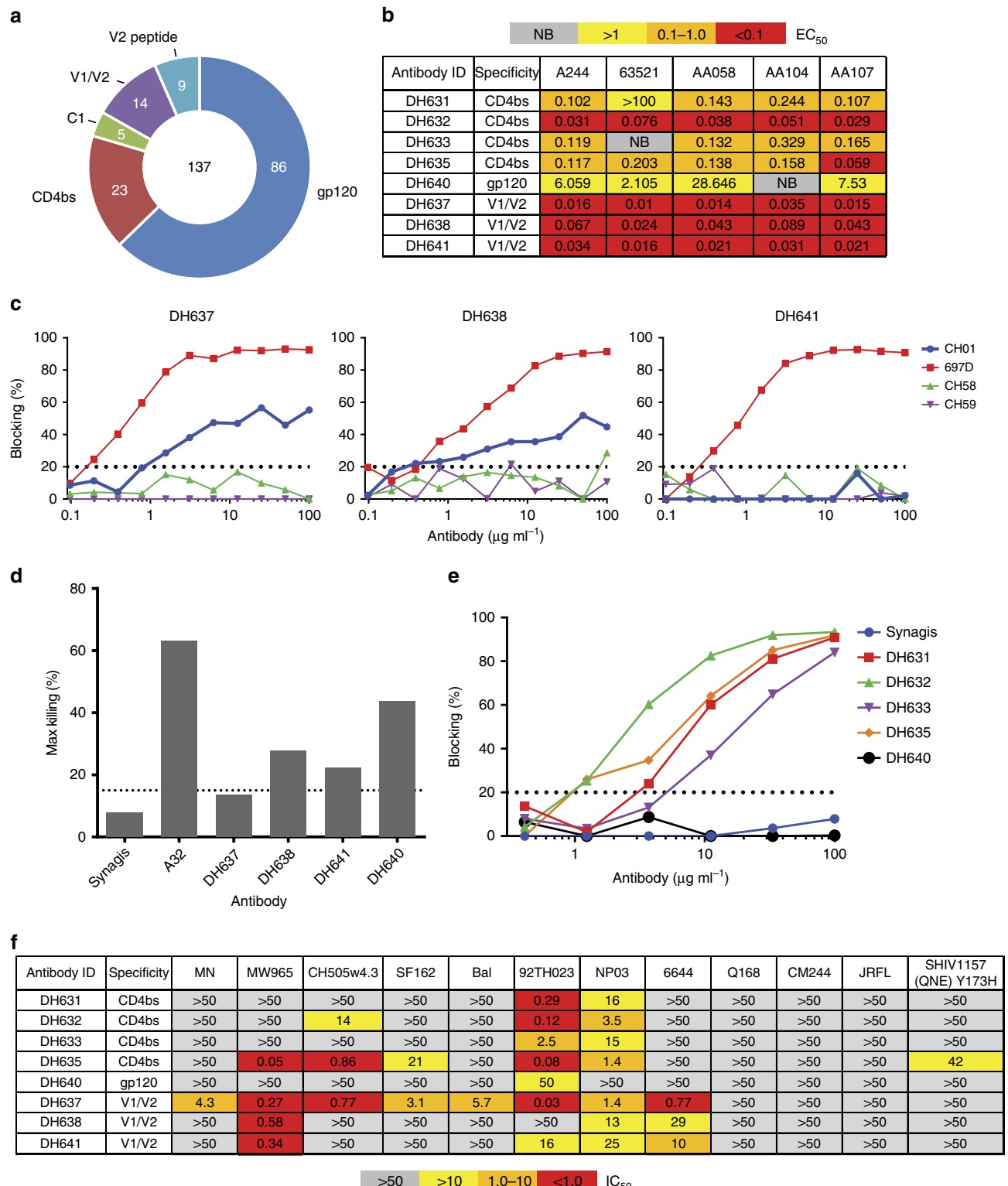

**Figure 6 | Isolation of antibodies with attributes of decreased infection risk.** (**a**) Chart of numbers of HIV-1-reactive mAbs isolated by antigen-specific single-cell sorting and tested for binding to gp120 proteins by ELISA. (**b**) Binding of eight mAbs to the five vaccine gp120 proteins in ELISA measured by EC$_{50}$. (**c**) Competitive blocking ELISA of CH01, 697D, CH58 and CH59 by isolated antibodies DH638, DH637 and DH641 measured as percentage of blocking. (**d**) ADCC of CM235-infected CD4$^+$ T cells by DH637, DH638, DH641 and DH640 measured by percentage of cell killing. Synagis and A32 antibodies used as negative and positive controls, respectively. (**e**) Competitive blocking ELISA of sCD4 by isolated antibodies DH631, DH632, DH633, DH635 and DH640 measured as percentage of blocking. (**f**) Antibody neutralization of Tier-1 viruses (MN, MW965, CH505w4.3, SF162, Bal, 92TH023, NP03 and 6644) and Tier-2 viruses (Q168, CM244 and SHIV-1157(QNE)Y173H) measured in the TZM-bl neutralization assay as IC$_{50}$.

protection[22,23], and previous immunization studies using adenovirus vectors and a pox-vectored or Env protein boost resulted in protection against a neutralization-resistant challenge SHIV-SF162P3 in non-human primates[24–26]. Study of humoral immune responses in previous rhesus macaque challenge studies suggested that improved protection is due to enhanced functional vaccine-elicited antibody responses to the HIV Env that work in combination. Identification of vaccine correlates of protection are of critical importance to inform future vaccine studies. Systems serology of antibody responses in our study identified that the bivalent and pentavalent vaccine arms elicited different antibody responses. The pentavalent-immunized animals had higher levels of antibodies that bound and mediated ADCP of AA104 and AA107 Envs that were only in the pentavalent vaccine and higher levels of antibodies that bound to the surface of HIV-1-infected CD4 cells. Multiple individual antibody parameters correlated with delayed infection, but a combination of three parameters that measure ADCC (antibody binding to HIV-infected cells, peak ADCC titres, NK cell-mediated ADCC) and a parameter that measures antibody-mediated NK cell activation (MIP-1β intracellular expression) were the best predictors of decreased infection risk. This indicated that improved ADCC, particularly HIV-infected cell killing mediated by NK cells, in the pentavalent-vaccinated animals resulted in increased protection from SHIV acquisition. Larger studies in primates and human will be required to replicate and confirm these immune correlates of protection.

While antibody responses to linear V2 peptides were higher in the pentavalent vaccine group, in the analysis of correlates of decreased transmission by Cox PH models, increased binding to AA104 and AA107 V2 peptides missed the significance threshold. Thus the exact epitope specificities of ADCC-mediating antibodies remains to be determined. However, it is important to note that we isolated both A32-blockable and V2 antibodies from RV144 vaccinees that synergized to mediate potent ADCC[27]. Moreover, we have isolated antibodies from pentavalent-immunized macaques that had broad binding to V2 epitopes and mediated ADCC or could inhibit the binding of α4β7 integrin. During acute HIV infection, CD4+ T cells that express high levels of the α4β7 integrin are infected, traffic the virus to gastrointestinal tissues and lead to formation of persistent viral reservoirs[28]. Recent studies have demonstrated that administration of α4β7-blocking mAbs can protect against SIV acquisition and can even result in sustained virological control after antiretroviral therapy[29,30]. In addition, we isolated a CD4bs antibody that could weakly neutralize the challenge SHIV. It is possible that low levels of the α4β7 blocking and SHIV-neutralizing antibodies could have had a protective effect. Our immune correlates studies suggest that non-neutralizing antibodies that mediate ADCC can afford protection against a Tier-2 SHIV in the absence of potent plasma Tier-2 neutralization similar to the observations in the human RV144 trial.

Finally, the present findings demonstrated that increasing a bivalent gp120 Env protein boost to a pentavalent gp120 Env protein boost that contained additional cross-reactive Env motifs improved antibody responses and resulted in decreased risk against acquisition of neutralization-resistant virus in rhesus macaques. Further studies will be required to determine whether one or all three additional Envs added to the pentavalent vaccine are required for the observed improved protection. These data have implications for future design of pox-vectored vaccine strategies and suggest that a polyvalent Env mixture may improve vaccine efficacy. Important differences exist between SHIV infection in rhesus macaques and HIV-1 infection in humans, including the limitation that we were only able to assess the

protective efficacy against a single SHIV in rhesus macaques. Further clinical efficacy studies in humans will be required to evaluate whether increasing Env diversity in a protein vaccine correlates with increased efficacy at preventing HIV-1 acquisition.

## Methods

**Phylogenetic tree.** The PHYML program was used[31] to create a phylogenetic tree based on the Env protein alignment from RV144 plus vaccine and challenge strains. The tree was constructed through the PHYML HIV database interface (www.hiv.lanl.gov) using an HIVb model, four rate categories and a gamma shape parameter of 0.31 estimated from the data. The tree was highlighted using Rainbow Tree (www.hiv.lanl.gov).

**Vaccine design and production.** Envelope sequences (1672) from 110 individuals that were infected during the RV144 study, including 44 RV144 vaccinees and 66 placebo recipients (ref. 11; GenBank accession numbers JX446645-JX448316) were aligned. All samples were collected within a 6-month window from infection. As we were interested in covering the overall CRF01_AE population diversity in Thailand, we used a consensus sequence to represent each sample, so each individual who was represented in our data set only once; the consensus was used as an approximation of the transmitted founder virus of each infection[32]. A full Env protein alignment containing these 110 within-subject consensus sequences, and also including all relevant vaccine strains and the SHIV challenge virus, is included as Supplementary Data (RV144-vaccine.fasta). The key V2 epitope region, between the hypervariable regions V1 and V2 (ref. 33; HXB2 positions 154–184) was then extracted from the alignment for vaccine antigen selection.

Three natural Env proteins (AA058, AA104 and AA107) were selected using the mosaic tool at the Los Alamos database (http://www.hiv.lanl.gov) with settings to select optimum viruses for coverage from among natural sequences in an alignment, not to make artificial mosaic proteins[15,34]. Proteins were selected based on coverage of contiguous 8-mers; we used 8-mers to capture local co-variation patterns. The Env consensus sequence from AA058, AA058.con, is identical to the natural sequence AA058a04R (JX448024); AA107.CON is identical to three natural Envs from AA107, including AA107.a_wg6 (JX448024); AA104.CON is closest to AA104.a_RH3 (JX447984) but one base off, so the consensus has an N, AA104.a_RH3 a D, at HXB2 position 611. B.63521 was chosen as the clade B component, rather than the MN Env used in RV144, based on superior antigenicity among tested transmitted/founder Envs[35].

The SHIV-1157(QNE)Y173H challenge virus was a clone of a heterologous clade C virus SHIV-1157ipd3N4 (ref. 14), which was mutated in three positions. Two mutations were introduced after a number of substitutions were tested to see what was needed to improve sensitivity to neutralization by V2 glycan antibodies. The mutations Q170K and I192R in combination provided greater sensitivity to PG9, PG16 and related quaternary-dependent neutralizing antibodies. 192R is a common amino acid in both the C clade and CRF01_AE, whereas I is very rare. The third mutation introduced was Y173H; this mutation was introduced because it enhanced reactivity with RV144 V2 antibodies CH58 and CH59.

Recombinant gp120 proteins B.63521, A244, AA058, AA104 and AA107 were produced as described[17,36]. HIV-1 Env V2 peptides (HXB2 positions 165–186; A244-Biotin-GGGGLRDKKQKVHALFYKLDIVPIED, AA104-Biotin-GGGIRDKKQ KAYALFYKLDLVPLKN, AA107-Biotin-GGGLKDKKQKVYALFYKLDIVPMPN, AA058-Biotin-GGGLRDKQQKVHALFYRLDIVPINS, B.63521-Biotin-GGGVRD KVQKEYALFYKLDIVPITN) were produced with an N-terminal biotin conjugated (CPC Scientific, San Jose, CA).

**Immunization and SHIV challenge of rhesus macaques.** Twenty-six Indian origin male and female rhesus monkeys (*Macaca mulatta*) were genotyped and selected as negative for the protective major histocompatibility complex class I allele *Mamu-A*01*. Monkeys were housed at New England Primate Research Center, Southborough, MA or Bioqual, Rockville, MD. The animals were maintained in accordance with the National Institutes of Health and Harvard Medical School guidelines and all studies were approved by the appropriate Institutional Animal Care and Use Committee.

Selected animals were randomly assigned to each vaccine arm. Monkeys were vaccinated by the intramuscular (IM) route with $1 \times 10^8$ pfu ALVAC (vCP1521; Sanofi Pasteur) vector alone twice and then immunized with ALVAC and 100 µg total purified Env gp120 protein in GLA-SE (IDRI-EM107) adjuvant with the exception of the last protein boost (week 88), which was 300 µg total protein in GLA-SE. ALVAC was delivered IM into one leg and the protein was delivered IM into the other leg. Animals were then challenged eight times weekly by the intrarectal route with 1:10,000 dilution of our SHIV-1157(QNE)Y173H challenge stock. The virus stock was grown from the infectious molecular clone in rhesus peripheral blood mononuclear cells (PBMCs) and the stock was titrated in rhesus

macaques to select the appropriate dilution. Sequence diversity of the viral stock was determined by single-genome amplification[37].

SHIV plasma viral RNA measurements were performed at the Immunology Virology Quality Assessment Center Laboratory Shared Resource, Duke Human Vaccine Institute, Durham, NC as described[37].

**Antibody binding.** Binding of vaccine plasma antibodies and mAbs to HIV-1 Envs was measured by ELISA in duplicate as described[38,39]. Binding and linear epitope mapping of plasma purified IgG or mAbs was performed by peptide microarray. A single microarray was performed for each animal[40,41]. Rectal mucosal IgG-binding responses were measured in duplicate by HIV-specific-binding antibody multiplex assays and total rhesus macaque custom ELISA. Specific activity was calculated by dividing the specific binding by the total IgG or IgA concentration. Positivity criteria were values threefold over the baseline visit and the cutoff was established using seronegative samples.

**Neutralization.** Neutralization activities of animal plasma and purified antibodies were determined by the TZM-bl-cell-based neutralization assay[42]. Tier phenotyping of the pseudoviruses was assayed by sensitivity to a pool of HIV-infected serum as described[43]. Neutralization assays are performed in technical triplicate for all animals at each serum time point or antibody concentration.

**ADCC against gp120-coated target cells.** The GranToxiLux assay was used to detect ADCC activities of NHP plasma samples directed against CEM.NKR$_{CCR5}$ CD4$^+$ T cells (NIH AIDS Reagent Program Division of AIDS, NIAID, NIH from Alexandra Trkola) coated with recombinant gp120 as described[44]. ADCC activities of fourfold serial plasma dilutions starting at 1:100 were measured against cells coated with gp120 isolates representing vaccine immunogens AA104, AA107 and A244 and the challenge virus, SHIV1157(QNE)Y173H. Cryopreserved human PBMCs from an HIV-seronegative donor (Duke University) with the heterozygous 158F/V genotype for Fc-gamma receptor IIIa were used as the source of effector cells[45]. Data were reported as the maximum proportion of cells positive for proteolytically active granzyme B out of the total viable target cell population (maximum %GzB activity) after subtracting the background activity observed in wells containing effector and target cells in the absence of plasma. ADCC end point titres (ADCC titre) were determined by interpolating the dilutions of plasma that intercept the previously established positive cutoff for this assay (8% GzB activity) using GraphPad Prism, version 6.0f software (GraphPad Software, Inc.). The percentage of NK-killed gp120-coated targets was evaluated using area scaling analysis of the GzB$^+$ cells. ADCC assays were performed in duplicate for each animal at each time point.

**ADCC against HIV-1-infected target cells.** The ability of NHP plasma samples to direct killing of CEM.NKR$_{CCR5}$ cells infected with HIV-1 isolate CM235 (GenBank accession no. AF259954.1, Agnès Chenine, US Military HIV Research Program) infectious molecular clone virus (IMC) containing a *Renilla* luciferase (Luc) reporter gene[46] was measured using a Luc-based ADCC assay according to a modified version of our previously described procedure[17,47]. Plasma was tested after fourfold serial dilutions starting at 1:100. Effector cells were the same PBMCs used for the GranToxiLux assay, stimulated overnight in the presence of media supplemented with 10 ng ml$^{-1}$ IL-15 to activate NK cells[48]. IL-15 was maintained in the media throughout the duration of the ADCC assay. Killing was measured as a reduction in luminescence (Viviren Assay, Promega) compared to that of control wells containing target and effector cells in the absence of plasma, and the final results (maximum percentage of specific killing and ADCC Ab titre, which is the last dilution of plasma above the previously established positive cutoff for this assay (15% specific killing) are reported after subtracting the baseline activity observed for samples collected prevaccination. ADCC assays were performed in duplicate for each animal at each time point.

**Plasma binding to the surface of HIV-1-infected cells.** Indirect surface staining was used to measure the ability of NHP plasma samples to bind HIV-1 envelope expressed on the surface of infected cells using methods similar to those previously described[19]. Briefly, mock-infected and CM235-IMC-infected CEM.NKR$_{CCR5}$ cells were incubated with 1:100 dilutions of NHP plasma samples for 2 h at 37 °C and then stained with a vital dye (Live/Dead Aqua) to exclude dead cells from analysis. The cells were then washed and permeabilized using BD Cytofix/Cytoperm solution. Cells were then washed again and stained with fluorescein isothiocyanate (FITC)-conjugated goat-anti Rhesus IgG (H + L) polyclonal antisera (Southern Biotech) to detect binding of the NHP plasma and RD1-conjugated anti-p24 (KC57, Beckman Coulter) to identify infected cells. Cells positive for NHP plasma binding were defined as live, p24 positive and FITC positive. Final results are reported as the percentage of FITC-positive cells and FITC MFI among the p24-positive events after subtracting the background observed for the prevaccination samples. Assays were performed in duplicate for each animal at each time point.

**Antibody-dependent cellular phagocytosis assay.** Antibody-dependent phagocytosis was assessed by the measurement of the uptake of antibody-opsonized, antigen-coated fluorescent beads by the monocytic THP-1 cell line (ATCC; #TIB-201)[49]. Briefly, THP-1 cells were purchased from ATCC and cultured as recommended. Antibody-mediated phagocytosis assay was performed as described[50], with the following modifications. Briefly, $9 \times 10^5$ beads (equivalent of 0.1 μl of supplied suspension) coupled with 63521, A244, AA104, AA107 or SHIV-1157(QNE)Y173H gp120 Env were mixed with 10 μl (25 μg ml$^{-1}$ final concentration) purified rhesus IgG in a 96-well round-bottom plate. After incubation at 37 °C for 2 h, $5 \times 10^4$ ( or $1.25 \times 10^4$ for SHIV-1157(QNE)Y173H to improve sensitivity) THP-1 cells were added to each well with final volume 200 μl each, then spinoculated at 1,200 g for 1 h at 4 °C. Blocking of CD4 on the cells was achieved by pretreating the cells at $10 \times 10^6$ cells ml$^{-1}$ with 20 μg ml$^{-1}$ anti-human CD4 antibody (clone SK3) (Biolegend) for 15 min at 4 °C before adding them to the antibody beads/virus mixture. Following spinoculation, antigens/viruses and cells were incubated at 37 °C for 1 h to initiate phagocytosis internalization. Cells were then fixed in 2% paraformaldehyde. Phagocytosis score was calculated by percentage of cells positive × mean MFI normalized to the corresponding result for the no-antibody control. A background level of phagocytosis was determined based on the mean + 3 s.d. of non-HIV-specific antibodies. Assays were performed in duplicate for each animal.

**Longitudinal antibody analysis.** A subset of assays (Supplementary Fig. 5B,C) were performed on samples from six time points (weeks 2, 6, 15, 23, 49 and 90; 2 weeks post each immunization) throughout the regimen as follows. Assays performed in technical duplicates for each animal.

**Fc array.** Fc and Fv characteristics of antigen-specific serum antibodies were evaluated using a custom, high-throughput, multiplexed array as described previously[26,51]. In brief, a panel of recombinant HIV and SHIV proteins were covalently coupled to fluorescent beads (Luminex). In duplicate, serum samples were diluted 1:1,000 into a 384-well microplate (Greiner Bio One) containing ~500 beads of each specificity per well. In addition to test samples, pooled positive and negative purified IgG samples were included. Beads were incubated in antibodies, washed and subsequently incubated with Fc detection reagents, including both rhesus[52] and human[53] FcgRs. Antigens and Fc detection reagents are listed in Supplementary Fig. 5C. Following incubation, plates were washed and data were acquired on a FlexMap3D instrument (Luminex). Data were reported as MFI values. Prior to analysis, Fc Array features were filtered for quality using a two-step process. First, features were excluded if the correlation between sample replicates did not exceed 0.7. Second, features whose average did not exhibit both a greater signal than a negative control sample and an MFI value of 500 were excluded.

**High-throughput Fc functional analysis.** IgG was purified from plasma samples using Melon Gel. All assays performed on all animals in duplicate and values reported are the average unless noted otherwise.

**Antibody-dependent neutrophil-mediated phagocytosis assay.** Antibody-dependent neutrophil-mediated phagocytosis was assessed by the measurement of the uptake of antibody-opsonized, antigen-coated fluorescent beads by primary neutrophils[26]. Biotinylated A244 gp120 was used to saturate the binding sites on 1 μm fluorescent neutravidin beads (ThermoFisher). Excess antigen was removed by washing the beads, which were then incubated with animal Ab samples for 20 min at 37 °C. Leukocytes were isolated from blood collected from HIV-seronegative donors by ACK lysis of red blood cells. Following opsonization, the freshly isolated leukocytes were added, and the cells were incubated for 1 h at 37 °C to allow phagocytosis. The cells were then stained for CD66b (BioLegend #305112; 1 μl per test) to identify neutrophils and fixed, and the extent of neutrophil phagocytosis was measured via flow cytometry (gating on CD66b-positive cell) on a Stratedigm S100EXi flow cytometer equipped with high-throughput sampler. The data are reported as a phagocytic score, which takes into account the proportion of effector cells that phagocytosed and the degree of phagocytosis (integrated MFI: frequency × MFI).

**Antibody-dependent complement deposition.** Ab-dependent complement deposition was assessed by the measurement of complement component C3b on the surface of target cells[26]. CD4-expressing target cells from healthy donors were pulsed with the A244 gp120 protein (6 μg per million cells) and incubated with purified Abs. Freshly isolated HIV-negative donor plasma diluted with veronal buffer and 0.1% gelatin (1:10 dilution) was added, and the cells were incubated for 20 min at 37 °C. The cells were then washed with 15 mM EDTA in PBS, and complement deposition was detected via flow cytometry following staining for C3b (Cedarlane #CL7632F; 1 μl per test). Replicates using heat-inactivated donor plasma were used as negative controls.

**Antibody-dependent NK cell activation.** Ab-dependent NK cell degranulation and cytokine/chemokine secretion was measured using freshly isolated NK cell responses to plate-bound A244 gp120 (ref. 54). Protein-binding plates were coated with A244 gp120 (300 ng per well) and incubated for 2 h at room temperature. Plates were then blocked in 5% BSA overnight at 4 °C. Fresh NK cells were isolated from whole blood from seronegative donors using negative selection with RosetteSep, as recommended by the manufacturer. Plates were washed to remove unbound antigen, and sample plasma was added and incubated for 2 h at 37 °C. Following the incubation, plates were washed to removed un-opsonized antibodies. Primary NK cells isolated from HIV-seronegative donors were incubated with anti-CD107a (BD #555802; 2.5 μl per test), brefeldin A (10 mg ml$^{-1}$) (Sigma) and GolgiStop (BD) for 5 h at 37 °C. The cells were then washed and stained for surface markers using anti-CD16 (BD #557758; 1 μl per test), anti-CD56 (BD #557747; 1 μl per test) and anti-CD3 (BD #558124; 1 μl per test). The cells were then washed, fixed and permeabilized using Fix & Perm (Invitrogen) and then stained intracellularly with anti-IFN-γ (BD #340449; 5 μl per test) and anti-MIP-1β (BD #550078; 1 μl per test). The cells were then fixed in 4% paraformaldehyde and analysed using flow cytometry. NK cells were defined as CD3-negative and CD16- and/or CD56-positive. All antibodies for flow cytometry were purchased from BD.

**Group classification.** Logistic regression models were developed to distinguish animals in the two groups according to antibody and functional measurements (Supplementary Fig. 5A–C). These models were trained using LASSO regularized logistic regression, seeking relatively sparse sets of measurements whose linear combinations differentiate the groups. The R package 'glmnet'[55] with default options was used to evaluate the effect of the regularization parameter (lambda), and models with lowest training error were selected. To analyse regression coefficients, a final model was trained using all the samples. To assess the overall accuracy of the antibody profile-based classification approach, we performed ninefold cross-validation, ensuring that one sample from each group was included in each testing set, and computed the balanced accuracy (mean true-positive rate) on testing set predictions. The robustness of this approach was evaluated by repeating the cross-validation over random training and testing set splits and by permutation testing (namely, repeating the same process with randomly shuffled group labels)[56]. Classification at each of the four time points after the initial boost (weeks 15, 23, 49 and 90) was performed using the measurements available at those time points.

**Survival analysis.** Models using antibody and functional measurements (Supplementary Fig. 5A,B) to predict risk of infection for each challenge were trained using Cox PH regression[57], as implemented in the R package 'survival'[58]. Due to the relatively large number of features compared to the number of samples, which can lead to overfitting of Cox models, aggressive feature-filtering approaches were employed, both in preprocessing and as part of the model training. Models were used to predict both animal-level and group-wise risk of infection, and as with group classification, overall model performance and robustness were assessed using repeated cross-validation and permutation testing. Representative models were trained with the most predictive features over the repetitions.

*Feature prefiltering.* Each feature's individual ability to predict risk was assessed in terms of a polyserial correlation coefficient[59], a metric estimating correlations between continuous and discrete variables. The top 25% of features ranked by coefficient magnitude were retained. In order to obtain a high-quality, non-redundant subset of these candidates, features were considered in decreasing order of polyserial coefficient magnitude. Subsequent features that were highly correlated (Pearson correlation coefficient > 0.8) to selected features were eliminated.

*Model training and feature selection.* For each fold in the cross-validation, a PH model was trained by selecting features using greedy backward elimination. First, an initial full model was trained on the complete set of filtered features. Then, for each feature, a separate model was trained excluding it, and the feature whose exclusion yielded the smallest drop in training set likelihood was eliminated. This procedure was repeated with successively reduced sets of features until either a 40% drop in likelihood compared to the initial full model was reached or only one feature was retained.

*Animal-level risk prediction and evaluation.* Since the PH model is semiparametric, with the baseline hazard function unspecified, the absolute risk of an individual animal cannot be estimated. However, its risk relative to the mean of all animals represents a single assessment that is invariant across the challenge time points. This relative risk then enables the evaluation of the model's performance with the C-index metric[60]. Higher C-indices indicate better agreement between the two variables, with a random level of agreement yielding a value of 0.5. As described for classification, risk was modelled with ninefold cross-validation, using 100 repetitions to estimate the variation in performance for different random splits of samples. Performance was compared with that of models trained the same way but using randomly shuffled challenge data, as in permutation testing.

*Representative models.* The most predictive features were selected as those appearing in at least 90% of the 900 PH models obtained over the 100 repetitions of ninefold cross-validation. These selected features were used to train representative models for a run of ninefold cross-validation to produce illustrative relative risk predictions for individual animals. Adding other features with lower frequencies did not improve the predictive power of the representative models.

*Group-wise risk prediction.* In addition to predicting the relative risk of infection of individual animals, a PH model can also be used to estimate the probability of infection at each challenge point for a test sample. For predicting group-wise risk, we trained a final model using all the samples, considering the most predictive features as discussed above for representative models. The survival probabilities for the bivalent and pentavalent groups were estimated by making predictions on 'mean' animals whose measurements represent mean values over the respective groups. The predicted and observed probabilities for each group were compared using the log-rank statistic. A high P value of the log-rank statistic provides stronger evidence for the null hypothesis that the two curves are similar.

**Antibody isolation.** A244 gp120, A244 V1/V2 tags and B.63521 V1/V2 tag proteins were tetramerized using fluorescently labelled streptavidin using AF647 (ThermoScientific) and BV421 (Biolegend). PBMCs from rhesus monkeys were stained with fluorescently labelled antibodies for cell surface markers (BD) and both fluorescently labelled proteins. Memory B cells (defined as viable singlet CD3 (BD #552852; 2.5 μl per test)/CD14 (BioLegend #301832; 5 μl per test)/CD16 (BD #557744; 5 μl per test) negative, CD20 (BD #347673; 5 μl per test) positive, surface IgD (Southern Biotech #2030-09; 1 μl per test) negative cells) that were stained doubly positive for A244 gp120, A244 V1/V2 tags or B.63521 V1/V2 tag proteins were sorted into single wells of 96-well PCR plates containing reverse transcriptase–PCR buffer as previously described[61]. Antibody variable heavy and variable light genes were amplified using nested PCR and purified and sequenced as described previously[61]. VDJ arrangements, clonal relatedness and identification of the intermediate and unmutated common ancestor were inferred using previously described computational methods[62,63].

**Expression of recombinant antibodies.** Transient small-scale expression of antibodies was achieved by assembling $V_H$, $V_K$ or $V_L$ sequences with linear cassettes that contain the cytomegalovirus promoter, respective Ig constant region and poly A signal sequence using overlapping PCR and transfection into 293T cells as described previously[64]. Supernatants were directly used to screen for binding of Env antigens in ELISA.

For production of purified recombinant mAbs, the $V_H$ and $V_L$ genes were synthesized (Genscript) and cloned into expression vectors and expressed and purified as described previously[65].

ELISA binding of transiently transfected supernatants and purified recombinant mAbs was performed as described previously[65].

**Inhibition of binding of α4β7 by mAbs.** Rhesus mAbs at a concentration of 20 or 50 μg ml$^{-1}$ in sample buffer were examined using our previously published α4β7 inhibition assay[66]. Briefly, triplicate wells of a 96-well plate were coated overnight at 4 °C with MAdCAM-1 (R&D systems #6056-MC-050; 2 μg ml$^{-1}$) or Streptavidin diluted in bicarbonate buffer. The streptavidin-coated plates were then incubated with biotinylated cyclic 92TH023-V2 peptide for 1 h at 37 °C. The plates were then blocked with blocking buffer for 1 h at 37 °C followed by the addition of rhesus mAb DH637 or control mAbs (Ab82 targets influenza and 7B2 targets the immunodominent region of gp41, all produced recombinantly at Duke) in sample buffer for 45 min at 37 °C. Plates were washed, followed by the addition of RPMI8866 cells (Sigma #95041316) that had been preincubated for 45 min at 37 °C with sample buffer or with sample buffer containing 0.5 μg ml$^{-1}$ of mAb ACT-1 (AIDS Research and Reference Reagent Program #11718; 5.7 μg ml$^{-1}$). Plates were then incubated for 1 h and washed and the adhered cells were detected by the addition of AlamarBlue dye. Fluorescence was measured using a M2 plate reader. Percentage of α4β7 inhibition was calculated by dividing the fluorescence units from cells incubated with media and mAbs from those incubated with media alone at the 8 h time point. Two independent experiments in triplicate were performed.

**Statistical analysis.** The numbers of NHPs for each group in this study have been determined in consultation with statisticians to be the minimum required for statistical significant analysis. Investigators were blinded to the vaccine arm of the samples at the time of assay but were not blinded when analysing the data. A one-tailed log-rank test was used to evaluate differences between group KM curves. A Cox PH model with time varying covariate was used for evaluating potential correlates of protection. Spearman correlations were used to evaluate a correlation between assay titre at week 90 and the number of weeks to infection. The PC Composite score is the first principle component of the measures that had a significant Spearman correlation with weeks to infection. The group comparison was performed by Exact Wilcoxon. Univariate comparisons were made using exact Wilcoxon–Mann–Whitney tests. All statistical analysis was performed using SAS v9.4 (SAS Institute, Inc., Cary, NC). GraphPad Prism version 6.01 was used for graphical representation.

**Ethics statement.** Rhesus macaques (*M. mulatta*) were housed at the New England Primate Research Center (Southborough, MA) or Bioqual, Inc. (Rockville, MD), in accordance with the standards of the American Association for

Accreditation of Laboratory Animal Care. The protocol was approved by Harvard Medical School's Institutional Animal care and Use Committee under protocol number 03503 and Bioqual's Institutional Animal care and Use Committee under OLAW Assurance Number A-3086-01. Harvard Medical School and Bioqual are IAAALAC accredited. This study was carried out in strict accordance with the recommendations in the Guide for the Care and Use of Laboratory Animals of the National Institutes of Health (NIH) and with the recommendations of the Weatherall report: 'The use of non-human primates in research'. All procedures were performed under anaesthesia using ketamine hydrochloride, and all efforts were made to minimize stress, improve housing conditions and to provide enrichment opportunities (for example, social housing when possible, objects to manipulate in cage, varied food supplements, foraging and task-oriented feeding methods, interaction with caregivers and research staff). Animals were killed by sodium pentobarbital injection in accordance with the recommendations of the panel on Euthanasia of the American Veterinary Medical Association.

Human PBMCs from HIV-1-negative individuals were collected with Institutional Review Board approval by the Duke Medicine Institutional Review Board for Clinical Investigations (Protocols Pro00006526, Pro00000873, Pro00009459) All subjects were consented following 45 CFR 46 and written informed consent was obtained by all participants. No minors were recruited into this study.

**Data availability**. The antibody sequences presented in this article have been submitted to GenBank (http://www.ncbi.nlm.nih.gov/genbank/) under accession numbers KY764307–KY764322. The data that support the findings of this study are available from the corresponding authors upon request.

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

# ARTICLE

47. Pollara, J. et al. Association of HIV-1 envelope-specific breast milk IgA responses with reduced risk of postnatal mother-to-child transmission of HIV-1. J. Virol. **89,** 9952–9961 (2015).

48. Carson, W. E. et al. Interleukin (IL) 15 is a novel cytokine that activates human natural killer cells via components of the IL-2 receptor. J. Exp. Med. **180,** 1395–1403 (1994).

49. Ackerman, M. E. et al. A robust, high-throughput assay to determine the phagocytic activity of clinical antibody samples. J. Immunol. Methods **366,** 8–19 (2011).

50. Tay, M. Z. et al. Antibody-mediated internalization of infectious HIV-1 virions differs among antibody isotypes and subclasses. PLoS Pathogens **12,** e1005817 (2016).

51. Brown, E. P. et al. High-throughput, multiplexed IgG subclassing of antigen-specific antibodies from clinical samples. J. Immunol. Methods **386,** 117–123 (2012).

52. Chan, Y. N. et al. IgG binding characteristics of rhesus macaque FcgammaR. J. Immunol. **197,** 2936–2947 (2016).

53. Boesch, A. W. et al. Highly parallel characterization of IgG Fc binding interactions. mAbs **6,** 915–927 (2014).

54. Chung, A. W. et al. Polyfunctional Fc-effector profiles mediated by IgG subclass selection distinguish RV144 and VAX003 vaccines. Sci. Transl. Med. **6,** 228ra238 (2014).

55. Friedman, J., Hastie, T. & Tibshirani, R. Regularization paths for generalized linear models via coordinate descent. J. Stat. Softw. **33,** 1–22 (2010).

56. Ojala, M. & Garriga, G. C. Permutation tests for studying classifier performance. J. Mach. Learn. Res. **11,** 1833–1863 (2010).

57. Cox, D. R. Regression models and life-tables. J. R. Stat. Soc. B **34,** 187–+ (1972).

58. Therneau, T. M. & Grambsch, P. M. Modeling Survival Data: Extending the Cox Model (Springer, 2000).

59. Drasgow, F. Polychoric and polyserial correlations. Encyclopedia Stat. **7,** 68–74 (1986).

60. Harrell, Jr F. E., Lee, K. L. & Mark, D. B. Multivariable prognostic models: issues in developing models, evaluating assumptions and adequacy, and measuring and reducing errors. Stat. Med. **15,** 361–387 (1996).

61. Wiehe, K. et al. Antibody light-chain-restricted recognition of the site of immune pressure in the RV144 HIV-1 vaccine trial is phylogenetically conserved. Immunity **41,** 909–918 (2014).

62. Kepler, T. B. Reconstructing a B-cell clonal lineage. I. Statistical inference of unobserved ancestors. F1000Res. **2,** 103 (2013).

63. Munshaw, S. & Kepler, T. B. SoDA2: a Hidden Markov Model approach for identification of immunoglobulin rearrangements. Bioinformatics **26,** 867–872 (2010).

64. Liao, H. X. et al. High-throughput isolation of immunoglobulin genes from single human B cells and expression as monoclonal antibodies. J. Virol. Methods **158,** 171–179 (2009).

65. Liao, H. X. et al. Initial antibodies binding to HIV-1 gp41 in acutely infected subjects are polyreactive and highly mutated. J. Exp. Med. **208,** 2237–2249 (2011).

66. Peachman, K. K. et al. Identification of new regions in HIV-1 gp120 variable 2 and 3 loops that bind to alpha4beta7 integrin receptor. PLoS ONE **10,** e0143895 (2015).

## Acknowledgements

We thank Thad Gurley, Lawrence Armand, Dawn J. Marshall, John Whitesides, Krissey E. Lloyd, Christina Stolarchuk, Cindy M. Bowman, Celia C. LaBranche, R. Whitney Edwards, Kaylan Whitaker, R. Glenn Overmann, Jessica Peel, Sam McMillan, David Beaumont, Derrick Goodman, Sheetal Sawant and Nicole Yates for expert technical assistance; Auguste Badiabo for assistance with statistical analysis; and Kelly Soderberg and Samantha Bowen for project management. This work was supported by the Center for HIV/AIDS Vaccine Immunology-Immunogen Discovery (Grant UMI-AI100645), a grant from the National Institutes of Health/National Institute of Allergy and Infectious Diseases/Division of AIDS, and a Collaboration for AIDS Vaccine Discovery Grant (Grant OPP1033098) from the Bill and Melinda Gates Foundation The views expressed are those of the authors and should not be construed to represent the positions of the US Army or the Department of Defense.

## Author contributions

T.B. performed assays, analysed and interpreted data and wrote and edited the manuscript. J.P., X.S., R.P., D.G., A.E., J.A.W., D.C.M., K.K.P., M.R., N.L.M., T.J.S., G.A., M.E.A., G.T. and G.F. designed, performed and analysed immunoassays. N.V., S.P. and C.B.-K. performed statistical analysis. S.S., H.B., L.M., R.S. and L.L.S. conducted animal studies. K.O.S. and H.-X.L. produced vaccine proteins. S.P. and J.T. designed and provided ALVAC. S.G.R. designed and provided vaccine adjuvant. S.-L.H., J.F.T. and A.P. designed challenge virus. T.B.K. performed antibody genetic analysis. M.A.M. designed and oversaw antibody isolation. B.T.K. designed and computationally selected vaccine components. B.F.H. designed the study, oversaw all experiments, analysed all data and wrote and edited the manuscript.

## Additional information

**Competing interests:** S.P. and J.T. are employees of Sanofi Pasteur. B.T.K. and B.F.H. have patent applications submitted on vaccine candidates used in this study. The remaining authors declare no competing financial interests.

