## [Peer Review File · Nature Communications]

Reviewers' comments:

Reviewer #1 (Remarks to the Author):

COMMENTS TO THE AUTHORS

This manuscript from the laboratory of Dr. Bart Haynes and his colleagues describes results of a study of a novel broad coverage env engineered pentavalent vaccine that appears to show significantly better protection than bivalent vaccine upon challenge with a SHIV construct that was engineered to include the amino acid changes in the V2 domain that enhanced reactivity to glycan dependent antibodies and mAbs CH58 and CH59. Overall, the rationale for this study appears to be to provide proof of concept that indeed the efficacy of the vaccine utilized in the RV144 trial can be markedly improved upon by including sequences that were broader in coverage and also describes several correlates of protection that may be useful to monitor vaccine candidates in general.

1. My understanding of the studies is as follows:

The authors sequenced viruses from the RV144 trial including those that were breakthroughs and those from the placebo group so that a wider coverage of env viral sequences present in the Thai population could be identified with a focus on the V1/V2 region because this was the region that appeared to be correlated with immune pressure in the initial RV144 trial. Analysis of the 8 a.a. linear sequence of the V2 region using the mosaic design tool led to the identification of 3 natural env sequences which were reasoned to provide the best coverage of the linear epitopes of the V2 region. The inclusion of additional strains in the protein boost immunogen, allowed the authors to broaden the coverage even more that would include the entire gp120. The authors immunized 2 groups of macaques (n = 9/group) six times with 100 million pfu of ALVAC-AE at 0, 4, 13, 21, 47, and 88 weeks and boosted group 1 with the bivalent (B/AE) env protein and group 2 with the pentavalent ((B/E/E/E/E) at weeks 13, 21, 47 and 88 in the GLA/SE adjuvant. Controls included 8 animals that were unvaccinated. The animals were then challenged intra-rectally starting at week 90 with low doses of the engineered SHIV-1157 (QNE) weekly for a total of 8 times. Plasma viremia showed that whereas 8/9 bivalent boosted animals got infected, only 4/9 pentavalent boosted animals got infected and 6/8 of the unvaccinated animals got infected. Statistical analyses showed no difference between the control unvaccinated and the pentavalent vaccine recipients but when the data from the unvaccinated and bivalent group were combined (14/17 got infected), there was statistical difference when compared with the pentavalent vaccinated group (4/9). There were no differences in the peak viral load or viral loads following set point. There clearly was difference in protection between the pentavalent and the bivalent recipients.

The authors then examined the immune responses of all these animals in attempts to identify correlates of protection in animals that received the pentavalent vaccine. The animals receiving the pentavalent vaccine appear to show increased antibody responses to both the gp120 and the V2 peptide specific responses of the AA104 and AA107 that is present in the penta vaccine. Similar levels of antibodies that blocked the ADCC inducing V2 localized antibody CH59 and the broadly NAb CH01 that targets the V2 region were found in the bi versus penta vaccinated animals suggesting that the inclusion of the penta was not required for the induction of such antibodies. Highlights of the antibody responses to linear peptides covering the entire gp160 of 7 clades and recombinants was the finding that

peptides within the V3 regions of the clade B and C strains were the dominant targets for sera from both group of animals (as has been noted previously). Basically whereas the penta vaccine group reacted to a higher number of peptides specially those that were uniquely present in the penta vaccine, the antibody responses were similar. Both groups also made similar levels of mucosal IgG antibody responses. Whereas there were no detectable NAb responses against the tier 2 SHIV, there were essentially similar NAb responses against tier 1 with penta groups having a higher responses to clade E tier 1 viruses as expected. Antibody mediated phagocytosis assay was also performed. In this assay, pre-challenge both groups had similar levels of antibodies against the vaccine coated targets, however, as expected, there were selective induction of phagocytosis functional Ab's against the AA104 and AA107 coated targets in the penta group. Antibody dependent complement deposition assays and antibody dependent NK cell activation was present in the sera from both groups albeit with somewhat higher titers in the penta group. Non-neutralizing ADCC activity data showed higher levels of antibodies that bound to virus infected cells in the penta group than the bivalent group. There was, however, no difference in peak ADCC titers in the 2 group of animals.

The authors then utilized a systems serology approach to profile the polyclonal antibody responses and included data for multiple measures of antibody binding, blocking, neutralization, ADCC, NK cell surface marker expression and ADCC titers at time of challenge in the 2 group of monkeys. In addition, Fc arrays were utilized as were the characterization of the Fc functions of the antibodies. Cross validation and permutations confirmed the robustness of such an approach. Antibody activity against only those antigens present in the pentavalent vaccine were shown to be the highest discriminators between the two groups and the antibody responses to these over time appear to show broader function.

Upon the finding that antibodies that blocked CD4 and had a variety of ADCC function correlated with delayed infection prompted the authors to isolate HIV reactive memory B cells and amplify the immunoglobulin receptor genes from 2 of the recipients of the pentavalent vaccine group. Of the 137 Ab's that showed binding to gp120, 51 showed reactivity to regions of the env of interest including CD4bs, C1, V1/V2 and V2 linear epitopes. Eight of these antibodies that target the CD4bs, V1/V2 were selected for more detailed study (by criteria that was not made clear!!!). Three of the V2 reactive antibodies reacted strongly with all 5 env's but interestingly all 3 failed to block the previously defined mAb's against V2 termed CH58 and CH59 that target the V2 that contain Lys169 suggesting an epitope that is distinct from these previously defined antibodies. Two of the 3 blocked the binding of CH01 denoting similar epitopes with one of them, D637, that bound to a linear epitope that contain the canonical a4b7 binding site involved in gut homing. Four of the mAb's showed significant ADCC activity against infected target cells. All 8 mAb's showed nAb against tier 1 and, except for 1, none showed any neutralizing activity against the tier 2 SHIV. The a4b7 site binding antibody D637 had the broadest neutralizing activity. In general all antibodies that bind to env gp120, mediate ADCC, block CH01 and CD4 binding showed high correlation with delayed time to infection.

The authors thus conclude that antibodies from animals that were recipients of the pentavalent vaccine in general had a high correlation with decreased infection risk.

2. STRENGTHS OF THIS MANUSCRIPT: This is clearly an outstanding piece of work by an

outstanding team of scientists. The major strengths lie in the incredibly detailed effort that the authors put forth in choosing the spectrum of immunogens. This was extremely carefully analyzed and chosen and the authors deserve a considerable amount of credit for building their vaccine platform and the challenge virus to make sure that proof of principle could be obtained. Equally impressive and absolutely remarkable were the extensive number of assays and statistical analyses that were performed by the authors. I believe their studies set a NEW benchmark for the type of analyses that need to be performed in such vaccine studies. To this reviewer it was completely overwhelming to even try and summarize their findings as described above. Their findings are without question highly valuable and sets forth a new paradigm for other labs to follow. The question being posed was outstanding, the way the study was executed was outstanding and the way the data were analyzed nothing short of outstanding.

3. WEAKNESSES: However, there were also a number of weaknesses that the authors should be able to address I believe. These include the following:

- a. It is not at all clear to this reviewer why the data from all 9 animals that received the pentavalent vaccine were lumped together for their analyses? This is difficult to understand and rationalize. Thus, it would seem that the authors should select data from the pentavalent vaccinated animals that got protected and compare those with the data from the pentavalent vaccinated animals that DID not get protected and then compare the data from the pentavalent vaccinated individuals that did not get protected with the bivalent and unvaccinated recipients. I realize that the numbers would be small but is that not what one should be aiming for to define distinguishing properties of the animals that were protected from those that were not? Perhaps an exercise in that direction may provide a more refined understanding of the "true" correlates of protection.
- b. The statement that the authors claim that for the first time the concept of broadening the vaccine immunogen leads to better protection needs to be modified. This is not in fact true as such an approach has been utilized by other labs previously in a number of other disease conditions. Along these lines, it should be made clear by the authors that they do not know whether one or more of their additions in the pentavalent vaccine was inducing the protective response. It could be just one of them.
- c. There were no data on MHC or other polymorphisms of the monkeys described. Were these animals selected for Mamu-A01, B08 and B17?
- d. Was the challenge SHIV1157 mutant stock titrated, because whereas the parent SHIV1157 was titrated, are the authors clear as to whether their newly reconstructed SHIV challenge stock underwent titration for selecting the dose utilized.
- d. Were the V2 monoclonal antibodies described screened for their ability to block the binding of anti-a4b7 and if so what were the results because these data would be very useful in the light of some recent studies on the anti-a4b7 mAb?
- e. Were gut biopsies performed to measure mucosal viral loads and if so, what were the data on pro-viral DNA levels in the gut tissues of the animals?
- f. Were T cell responses measured in these animals and if so can a brief statement be made?
- g. The first paragraph of the discussion section is recommended for deletion since this is a repeat of the paragraph in the introduction. The authors should in fact take some time to discuss their approach of using broadened sets of immunogens for effecting a much broader

immune response.

Reviewer #2 (Remarks to the Author):

This work includes new findings that represent an advance in the HIV vaccine field. The results suggest that increasing the valency of an HIV Env vaccine with differences in key regions that were associated with decreased risk of infection of the RV144 trial may help increase the breadth of the immune response against more viral isolates and increase protection against infection.

While some of the findings were not statistically significant, the data trended in a complementary fashion across multiple different assays and the thorough array of testing strengthens the validity of the results.

While the manuscript is well written, well thought out, and easy to read for someone in the HIV vaccine field, limiting the use of field-specific jargon and defining unique terms at first use could help make this manuscript more accessible to a broader scientific audience.

Specific minor comments include the following:

Page 3, Line 63: minor comment, moving up the definition of the pentavalent B/E/E/E/E construct in lines 68-69 to line 63 would make it clearer to those new to what is meant by B/E/E/E/E right away.

Page 3, Line 58: ALVAC-HIV vCP1521 expresses Gag and Pro, the insert does not include the full length pol

Page 3, Lines 6: a very brief descriptor of what is meant by variation in "other gp120 regions" would be helpful up front.

Page 6, line 117: The use of the word "significantly" is misleading. The ALVAC-pentavalent vaccine did not afford statistically significant better protection against acquisition compared to the ALVAC-bivalent vaccine regimen. There was a trend. The significant difference was only observed when the bivalent and control groups were combined.

The rationale for combining the control group with the bivalent group to compare against the pentavalent vaccine group should be provided. Typically, this combination would not be a strong comparison. One would expect that either test arm may be better than the control group. The challenge results for this study, however, indicate that the bivalent group did no better than the control group in protection.

Page 14, line 301: The authors state, "limited protection (11.1%) with an ALVAC-bivalent protein regimen", but this level was actually lower than the control group. Suggest deleting the words "limited protection".

Page 14, line 316: minor typo "these types neutralizing antibodies" should be "these types of neutralizing antibodies"

Page 16, lines 349-354: Why focus on the pox-vector vaccine component as the main conclusion when the results were due to the protein boost? It is feasible that a polyvalent approach may be beneficial regardless of the specific vector.

Page 17, lines 388-391: V2 was assessed thoroughly, but what do the authors think is the relative importance and effect of additional V2 representation vs the differences in other portions of the Envs?

Page 21, lines 466, 468: typo, "9 x 10⁵ beads" should be 9 x 10⁵ beads and "cells at 10 x10⁶" should be cells at 10 x 10⁶.

Page 24, line 541: typo "incubated"

Reviewer #3 (Remarks to the Author):

The manuscript by Bradley et al describes the design of an ALVAC-pentavalent clade B/E/E/E/E envelope (Env) based vaccine that increases the diversity of the Env gp120 motifs compared a bivalent clade B/E Env based vaccine similar to the one used in in the RV144 vaccine trial. The ALVAC-pentavalent gp120 immunogen was designed to have the potential to induce broader responses to gp120 V2 and other gp120 epitopes compared to the ALVAC-bivalent gp120 immunogen. The pentavalent vaccine did induce a broader antibody (Ab) response and higher Ab titres directed to pentavalent-vaccine matched gp120 proteins, including those directed at V2 epitopes. A higher frequency of macaques receiving the pentavalent than the bivalent vaccine (55% vs 11%) were protected from multiple low dose challenges with a heterologous neutralization resistant (tier 2) SHIV. Immune correlates analyses were performed and showed neither vaccine was able to induce neutralizing Abs to the challenge virus.

Systems serology approaches showed that the two vaccine arms elicited different Ab responses. The animals receiving pentavalent vaccine had higher Ab levels that bound and mediated ADCC to 2 of the gp120 present only in the pentavalent vaccine arm and higher levels of Abs that bound to the surface of infected CD4 cells. Several Ab parameters correlated with delayed infection. A grouping of 3 parameters that measure aspects of ADCC (Ab binding to HIV-infected cells, peak ADCC titers, NK cell-mediated ADCC) and a parameter that measures Ab dependent NK cell activation (MIP-1 β expression) were the best predictors of reduced infection risk.

The results presented support the main conclusions made by the authors that designing vaccines to increase coverage of induced responses can improve vaccine induced protection. The use of systems serology approaches is a powerful tool for the analyses of immune correlates of protection in such vaccine trials.

Critiques

Overall, this is a well-designed and conducted study. The results are provide novel insights that build upon the modest success of the RV144 vaccine trial. They bring the field a step closer to the design of protective vaccines for HIV.

For the competitive ELISA experiments shown in Figure 2D, please add to the text or the legend for this figure from what time point the vaccine plasma samples came from used in this assay. From the results shown in Figure S2A, at least for the pre-challenge 90 week time point, it would appear that there would be potential differences in the ability of plasma from the 2 vaccine recipient arms to block binding of CH58, CH01, sCD4, but not A32, to A244 gp120. This is not what the statistics show in Figure 2D. Please clarify this point. Only blocking of A244 gp120 is reported. Were competitive ELISAs also performed using the other gp120s? If so, how did results compare to those for A244?

Figure 3C-G show results using plasma IgG from vaccine recipients as the source of Ab for ADCD, ADNP and Ab dependent NK cell activation measured by IFN- γ and MIP-1 β secretion and CD107a expression. The legends for these panels need to be modified to clarify what stimulatory/target CD4+ cell was used for these experiments. It appears that the gp120 used to coat target cells was A244. Since binding and ADCP experiments reported in Figures 2C, 2D and 3D show that AA104 and AA107 gp120s were recognized better by Abs from the recipients of pentavalent than bivalent vaccines were these experiments also done using targets coated with these gp120s?

Please specify in the legends for Figure 3I, 3J and S3C what ADCC assays was performed to generate these results. There are 2 described in the methods section, the GTL assay and the RFADCC assay. Also 2 targets are described in the methods section for the GTL assay, i.e. CEM.NKr cells coated with gp120 and CEM.NKr infected with HIV. The manuscript sections reporting the results of these ADCC experiments should specify which variations of these assays was used. Using the term CD4+ to describe these targets does not provide sufficient information. Also, what is meant by peak ADCC titre is not clear. This should be better described in the methods sections and/or the legend for panels 3J and S3C. This is important as peak ADCC titre arises as one of the important correlates for protection.

Please check the labelling for Figure S5F. There appear to be 2 panels labelled Fcg2AR.AA104 gp120 and 2 labelled Fcg2AR.AA058.

In line 236 and 331, do the authors mean intracellular MIP-1 β rather than surface MIP-1 β ?

In lines 501 and 511 patient is used when I think animal or macaque is meant. In line 512, HIV-seronegative is used when SIV in meant.

The following lines have types or dropped words.

Line 228

Line 246

Line 383 insert than after rather.

Reviewer #4 (Remarks to the Author):

The RV144 trial has provided evidence that a vaccine that includes the pox vector ALAC and a bivalent Clade B/E envelope was successful in protecting human subjects from infection by HIV. In this manuscript the author show the superiority of a similar pox vector based vaccine that also included pentavalent gp120 HIV recombinant proteins in protecting NHPs from SHIV infection. The authors modeled the envelopes on the sequence of breakthrough envelopes from the Thai trial. The authors go on a lengthy process of characterizing the immune response to the envelope moiety of the vaccine. They show as would be expected that animals immunized with the pentavalent env vaccine have a broader coverage of env epitopes. The magnitude and durability of the Ab response was similar in both arms. The authors go on to identify correlates of immune mediated protection that can distinguish animals immunized with each vaccine. Figure 3 show that both vaccines elicit similar magnitudes of Abs with effector functions. Figure 4 shows in contrast to figure 3 that Fcg binding capacity of Abs generated by each vaccine is different with the pentavalent vaccine showing higher magnitude in Fc receptor binding. Correlates of protection showed that peak ADCC titers were the immunological parameter that best correlated with protection. They identify in a very rigorous analysis that a composite score of other immune functions also can predict increased protection specifically in animals immunized with the pentavalent vaccine. They isolate Abs from protected macaques and show that they share features of plasma Abs associated to protection.

The work presented in this manuscript is comprehensive and very carefully performed. It shows the features of a vaccine that is modification of the RV144 vaccine. The authors perform an elaborate set of studies to identify the immunological mechanisms that underlie the enhanced protection observed in NHPs immunized with the pentavalent vaccine. Conclusions of these studies are straightforward and important as adding more envelopes to the vaccine can lead to increased protection. Having identified correlates of this enhanced protection will facilitate the development of this vaccine for human trials. This is an important study and should be published.

A minor suggestion : The authors should focus on presenting in the main body of the paper only those results that definitely differentiate the two vaccines and remove from the main body of the paper those features which are not discriminatory

Revised assay data included in manuscript: In a new ADCP assay we utilized a lower cell density to increase assay sensitivity for the assay with the challenge SHIV. This demonstrated differences between the two vaccine groups that correlated with delayed infection. However, this assay measure did not result in one of the 4 best predictors of delayed infection risk in our systems analysis. We included the new assay measurements in Fig. 3C. and the individual ADCP correlation in Fig. 5F. We modified the ADCP methods section to include these revised methods.

Response to reviewers comments:

Reviewer #1

1. Eight of these antibodies that target the CD4bs, V1/V2 were selected for more detailed study (by criteria that was not made clear).

RESPONSE: We selected the antibodies that had the broadest binding to the 5 vaccine immunogens. We have now included “and had the broadest binding to the vaccine immunogens” on line Page 12, line 265.

2. It is not at all clear to this reviewer why the data from all 9 animals that received the pentavalent vaccine were lumped together for their analyses? This is difficult to understand and rationalize. Thus, it would seem that the authors should select data from the pentavalent vaccinated animals that got protected and compare those with the data from the pentavalent vaccinated animals that DID not get protected and then compare the data from the pentavalent vaccinated individuals that did not get protected with the bivalent and unvaccinated recipients. I realize that the numbers would be small but is that not what one should be aiming for to define distinguishing properties of the animals that were protected from those that were not? Perhaps an exercise in that direction may provide a more refined understanding of the “true” correlates of protection.

RESPONSE: In our analysis, we aimed to define correlates of delayed infection regardless of the vaccine group. The correlates of delayed infection are not exclusive to the pentavalent group but were induced at higher levels in the pentavalent immunized animals. In both Figures 4 and 5, we colored the individual animals by vaccine group to show the group differences. Although, with a larger sample size of pentavalent animals the analysis the reviewer suggested would be worthwhile, comparing the 4 animals from the pentavalent group that got infected with the 5 that were protected would not be powered enough to demonstrate statistical differences.

3. The statement that the authors claim that for the first time the concept of broadening the vaccine immunogen leads to better protection needs to be modified. This is not in fact true as such an approach has been utilized by other labs previously in a number of other disease conditions. Along these lines, it should be made clear by the authors that they do not know whether one or more of their additions in the pentavalent vaccine was inducing the protective response. It could be just one of them.

RESPONSE: We did not intend to imply that our study was the first demonstration of improving vaccine efficacy by broadening the immunogens. We have now deleted that claim.

We agree that future studies will be required to determine which of the three additional Envs are required for improved protection. We have added the following to page 15 of the discussion: “Further studies will be required to determine if one or all three additional Envs added to the pentavalent vaccine are required for the observed improved protection.”

4. There were no data on MHC or other polymorphisms of the monkeys described. Were these animals selected for Mamu-A01, B08 and B17?

RESPONSE: Yes, the animals were screened for Mamu-A01 and selected if negative. This statement is in the materials and methods on line 399.

5. Was the challenge SHIV1157 mutant stock titrated, because whereas the parent SHIV1157 was titrated, are the authors clear as to whether their newly reconstructed SHIV challenge stock underwent titration for selecting the dose utilized.

RESPONSE: Yes, the SHIV was titrated to select the dose and we have added this statement on line 410 in the materials and methods to clarify.

6. Were the V2 monoclonal antibodies described screened for their ability to block the binding of anti-a4b7 and if so what were the results because these data would be very useful in the light of some recent studies on the anti-a4b7 mAb?

RESPONSE: Yes, the V2 antibody did block the binding of a4b7. We have now included these results Figure S7E. Additionally, we now added these results in the discussion section, and have added Mangala Rao and Kristina Peachman as authors who performed this assay.

7. Were gut biopsies performed to measure mucosal viral loads and if so, what were the data on pro-viral DNA levels in the gut tissues of the animals?

RESPONSE: Unfortunately, we did not collect gut biopsies for this study. That is a helpful suggestion and we will incorporate that into any future follow-up studies.

8. Were T cell responses measured in these animals and if so can a brief statement be made?

RESPONSE: For this study, we set out to measure protective antibody responses to follow along with the correlates of protection observed in the RV144 human trial. We did not measure T cell responses.

g. The first paragraph of the discussion section is recommended for deletion since this is a repeat of the paragraph in the introduction. The authors should in fact take some time to discuss their approach of using broadened sets of immunogens for effecting a much broader immune

response.

RESPONSE: We have deleted the first paragraph of the discussion section. (Page 13).

Reviewer #2

1. While the manuscript is well written, well thought out, and easy to read for someone in the HIV vaccine field, limiting the use of field-specific jargon and defining unique terms at first use could help make this manuscript more accessible to a broader scientific audience.

RESPONSE: We have revised the manuscript to better define unique terms at first use, including in the abstract.

2. Specific minor comments include the following:

Page 3, Line 63: minor comment, moving up the definition of the pentavalent B/E/E/E/E construct in lines 68-69 to line 63 would make it clearer to those new to what is meant by B/E/E/E/E right away.

RESPONSE: We have made this correction.

3. Page 3, Line 58: ALVAC-HIV vCP1521 expresses Gag and Pro, the insert does not include the full length pol

RESPONSE: We have made this correction. We labeled Pol by mistake.

4. Page 3, Lines 6: a very brief descriptor of what is meant by variation in “other gp120 regions” would be helpful up front.

RESPONSE: We have added a descriptor to mean gp120 epitopes outside of the V2 loop.

5. Page 6, line 117: The use of the word “significantly” is misleading. The ALVAC-pentavalent vaccine did not afford statistically significant better protection against acquisition compared to the ALVAC-bivalent vaccine regimen. There was a trend. The significant difference was only observed when the bivalent and control groups were combined.

The rationale for combining the control group with the bivalent group to compare against the pentavalent vaccine group should be provided. Typically, this combination would not be a strong comparison. One would expect that either test arm may be better than the control group. The challenge results for this study, however, indicate that the bivalent group did no better than the control group in protection.

RESPONSE: When the bivalent and pentavalent arms were compared there was a significant difference (Line 117 $p = 0.02$). There was not a significant difference between the pentavalent compared with controls alone (Line 119, $p = 0.15$). However, when we combine the bivalent and control animals the pentavalent group is significant again (Line 122 $p = 0.03$). As the reviewer suggests, we hypothesized that the bivalent group did no better than the controls and so we group them with the controls for an additional comparison of significant. However, even comparing the pentavalent with the bivalent

animals alone there was significantly better protection. We have added the sentence on line 118 to clarify.

6. Page 14, line 301: The authors state, “limited protection (11.1%) with an ALVAC-bivalent protein regimen”, but this level was actually lower than the control group. Suggest deleting the words “limited protection”.

RESPONSE: We have made this correction.

7. Page 14, line 316: minor typo “these types neutralizing antibodies” should be “these types of neutralizing antibodies”

RESPONSE: We have made this correction.

8. Page 16, lines 349-354: Why focus on the pox-vector vaccine component as the main conclusion when the results were due to the protein boost? It is feasible that a polyvalent approach may be beneficial regardless of the specific vector.

RESPONSE: We agree and have removed the pox-vectored component.

9. Page 17, lines 388-391: V2 was assessed thoroughly, but what do the authors think is the relative importance and effect of additional V2 representation vs the differences in other portions of the Envs?

RESPONSE: Our initial hypothesis was based on the immune correlate of V2 antibodies observed in the RV144 human trial. In our study while ADCC and NK cell activation correlated with delayed infection the V2 antibody response did not. This raises the hypothesis that other epitopes outside of the V2 were critical for protection. Future studies are underway to further identify specific protective epitopes.

10. Page 21, lines 466, 468: typo, “9 x 10⁵ beads” should be 9 x 10⁵ beads and “cells at 10 x10⁶” should be cells at 10 x 10⁶.

RESPONSE: We have made this correction.

11. Page 24, line 541: typo “incubated”

RESPONSE: We have made this correction.

Reviewer #3

1. For the competitive ELISA experiments shown in Figure 2D, please add to the text or the legend for this figure from what time point the vaccine plasma samples came from used in this assay.

RESPONSE: We added that these were assayed at week 90 to the figure legend.

2. From the results shown in Figure S2A, at least for the pre-challenge 90 week time point, it would appear that there would be potential differences in the ability of plasma from the 2 vaccine recipient arms to block binding of CH58, CH01, sCD4, but not A32, to A244 gp120. This is not what the statistics show in Figure 2D. Please clarify this point.

RESPONSE: In Figure S2A, the data are the mean of each group graphed. We have now added that to the legend. The results in Figure 2D show all the individual animals. While CH59, CH01 have different averages these were not statistically significant. The variation among the animals for the A32 blocking assay was much lower so there was a smaller average difference when compared to the other assays, but still achieved statistical significance.

3. Only blocking of A244 gp120 is reported. Were competitive ELISAs also performed using the other gp120s? If so, how did results compare to those for A244?

RESPONSE: Only A244 was utilized for competitive ELISAs for all antibodies. Soluble CD4 blocking was performed on other gp120s and produced similar results so we displayed A244 results for consistency.

4. Figure 3C-G show results using plasma IgG from vaccine recipients as the source of Ab for ADCD, ADNP and Ab dependent NK cell activation measured by IFN- γ and MIP-1 β secretion and CD107a expression. The legends for these panels need to be modified to clarify what stimulatory/target CD4+ cell was used for these experiments. It appears that the gp120 used to coat target cells was A244. Since binding and ADCP experiments reported in Figures 2C, 2D and 3D show that AA104 and AA107 gp120s were recognized better by Abs from the recipients of pentavalent than bivalent vaccines were these experiments also done using targets coated with these gp120s?

RESPONSE: We have added the target cells to the figure legends for all of the assays. The high-throughput Fc functional analysis was initially performed only with A244 due to the number of assays being performed.

5. Please specify in the legends for Figure 3I, 3J and S3C what ADCC assays was performed to generate these results. There are 2 described in the methods section, the GTL assay and the RFADCC assay. Also 2 targets are described in the methods section for the GTL assay, i.e. CEM.NKr cells coated with gp120 and CEM.NKr infected with HIV. The manuscript sections reporting the results of these ADCC experiments should specify which variations of these assays was used. Using the term CD4+ to describe these targets does not provide sufficient information. Also, what is meant by peak ADCC titre is not clear. This should be better described in the methods sections and/or the legend for panels 3J and S3C. This is important as peak ADCC titre arises as one of the important correlates for protection.

RESPONSE: We thank the reviewer for pointing out an error in the figure legend of figure 3. The RFADCC assay was not performed to generate these results. We have removed this

from the methods and corrected the figure legend with the correct assay and target cells used. We have also now defined ADCC titer in the figure legends and ADCC methods. It is the endpoint titer of positivity in the ADCC assay.

6. Please check the labelling for Figure S5F. There appear to be 2 panels labelled Fcg2AR.AA104 gp120 and 2 labelled Fcg2AR.AA058.

RESPONSE: The bottom two panels were labeled incorrectly and we have now corrected.

7. In line 236 and 331, do the authors mean intracellular MIP-1 β rather than surface MIP-1 β ?

RESPONSE: Yes, we meant surface MIP-1 β and we have now corrected.

8. In lines 501 and 511 patient is used when I think animal or macaque is meant. In line 512, HIV-seronegative is used when SIV in meant.

RESPONSE: We have corrected the patient error. The HIV-seronegative donors were used as effector cells in these assays not rhesus macaque.

9. The following lines have types or dropped words.

Line 228

RESPONSE: We have made this correction.

Line 246

RESPONSE: We have made this correction.

Line 383 insert than after rather.

RESPONSE: We have made this correction.

Reviewer #4

1. A minor suggestion : The authors should focus on presenting in the main body of the paper only those results that definitely differentiate the two vaccines and remove from the main body of the paper those features which are not discriminatory

RESPONSE: We have now removed Fig 3 panels C and D to the supplement as ADCD and ADNP were not different between the groups and did not have any correlation with delayed infection risk. We left some figures in the main text that show trends towards differences since they may be important for comparison with future studies.

REVIEWERS' COMMENTS:

Reviewer #1 (Remarks to the Author):

I believe the authors have addressed all the issues that I raised and I am very satisfied by their response. I have no additional comments with regards to this manuscript.

Reviewer #2 (Remarks to the Author):

The authors have adequately addressed comments, and the manuscript is suitable for publication.

Reviewer #3 (Remarks to the Author):

The authors have responded to reviewer comments and changes the manuscript to address reviewer critiques. I have no further changes to request.